# Schnyder corneal dystrophy-associated UBIAD1 inhibits ER-associated degradation of HMG CoA reductase in mice

Youngah Jo[1], Jason S Hamilton[1], Seonghwan Hwang[1], Kristina Garland[1], Gennipher A Smith[1], Shan Su[1], Iris Fuentes[1], Sudha Neelam[2], Bonne M Thompson[3], Jeffrey G McDonald[3], Russell A DeBose-Boyd[1]*

[1]Departments of Molecular Genetics, Center for Human Nutrition, University of Texas Southwestern Medical Center, Dallas, United States; [2]Department of Ophthalmology, Center for Human Nutrition, University of Texas Southwestern Medical Center, Dallas, United States; [3]Center for Human Nutrition, University of Texas Southwestern Medical Center, Dallas, United States

**Abstract** Autosomal-dominant Schnyder corneal dystrophy (SCD) is characterized by corneal opacification owing to overaccumulation of cholesterol. SCD is caused by mutations in UBIAD1, which utilizes geranylgeranyl pyrophosphate (GGpp) to synthesize vitamin $K_2$. Using cultured cells, we previously showed that sterols trigger binding of UBIAD1 to the cholesterol biosynthetic enzyme HMG CoA reductase (HMGCR), thereby inhibiting its endoplasmic reticulum (ER)-associated degradation (ERAD) (Schumacher et al. 2015). GGpp triggers release of UBIAD1 from HMGCR, allowing maximal ERAD and ER-to-Golgi transport of UBIAD1. SCD-associated UBIAD1 resists GGpp-induced release and is sequestered in ER to inhibit ERAD. We now report knockin mice expressing SCD-associated UBIAD1 accumulate HMGCR in several tissues resulting from ER sequestration of mutant UBIAD1 and inhibition of HMGCR ERAD. Corneas from aged knockin mice exhibit signs of opacification and sterol overaccumulation. These results establish the physiological significance of UBIAD1 in cholesterol homeostasis and indicate inhibition of HMGCR ERAD contributes to SCD pathogenesis.
DOI: https://doi.org/10.7554/eLife.44396.001

*For correspondence:
Russell.Debose-Boyd@
utsouthwestern.edu

Competing interests: The authors declare that no competing interests exist.

## Introduction

Mutations in the gene encoding UbiA prenyltransferase domain-containing protein-1 (UBIAD1) cause Schnyder corneal dystrophy (SCD), a rare autosomal dominant eye disease characterized by opacification of the cornea (*Klintworth, 2009*; *Weiss, 2009*; *Orr et al., 2007*; *Weiss et al., 2007*). Although apparent early in life, corneal opacification associated with SCD progresses slowly with age and ultimately leads to reduced visual acuity. The severity of visual impairment is underscored by the frequency in which corneal transplant surgery is utilized for treatment of SCD, which rises from 50% in SCD patients > 50 years of age to more than 70% for those 70 years and older (*Weiss, 2007*). Biochemical analyses of corneas from SCD patients revealed a marked accumulation of cholesterol (*McCarthy et al., 1994*; *Gaynor et al., 1996*; *Yamada et al., 1998*), suggesting that dysregulation of cholesterol metabolism significantly contributes to the pathogenesis of SCD. Systemic hypercholesterolemia has been reported to be associated with some, but not all cases of the disease (*Thiel et al., 1977*; *Brownstein et al., 1991*; *Crispin, 2002*).

UBIAD1 belongs to the UbiA superfamily of integral membrane prenyltransferases that catalyze transfer of isoprenyl groups to aromatic acceptors, generating a wide variety of molecules ranging from ubiquinones, chlorophylls, and hemes to vitamin E and vitamin K (*Li, 2016*). UBIAD1 mediates transfer of the 20-carbon geranylgeranyl moiety from geranylgeranyl pyrophosphate (GGpp) to menadione (vitamin $K_3$) released from plant-derived phylloquinone (vitamin $K_1$), generating the vitamin $K_2$ subtype menaquinone-4 (MK-4) (*Nakagawa et al., 2010*; *Hirota et al., 2013*). To date, 25 missense mutations that alter 21 amino acids in the UBIAD1 protein have been identified in SCD families. Structural analyses of archaeal UbiA prenyltransferases revealed that residues corresponding to SCD-associated mutations in human UBIAD1 cluster around the active site of the enzyme (*Cheng and Li, 2014*; *Huang et al., 2014*). Indeed, all SCD-associated variants of UBIAD1 are defective in mediating synthesis of MK-4 (*Hirota et al., 2015*; Jun, D.-J. and DeBose-Boyd, R.A., unpublished observations) and for some variants, this defect likely results from reduced affinity for GGpp.

The first link between UBIAD1 and cholesterol metabolism was provided by the discovery of its association with the endoplasmic reticulum (ER)-localized enzyme 3-hydroxy-3-methylglutaryl coenzyme A reductase (HMGCR) (*Nickerson et al., 2013*). HMGCR catalyzes reduction of HMG CoA to mevalonate, a reaction that constitutes the rate-limiting step in synthesis of cholesterol and the essential nonsterol isoprenoids GGpp and farnesyl pyrophosphate (Fpp) (*Goldstein and Brown, 1990*; *Wang and Casey, 2016*). Fpp and GGpp can become transferred to many cellular proteins and are utilized in synthesis of other nonsterol isoprenoids including ubiquinone, heme, dolichol, and MK-4. HMGCR is subjected to tight feedback control through transcriptional and post-transcriptional mechanisms mediated by sterol and nonsterol isoprenoids (*Brown and Goldstein, 1980*). Sterols mediate the transcriptional effects by inhibiting proteolytic activation of membrane-bound transcription factors called sterol regulatory element-binding proteins (SREBPs). SREBPs enhance transcription of genes encoding HMGCR and other cholesterol biosynthetic enzymes as well as the low density lipoprotein (LDL) receptor that removes cholesterol-rich LDL from circulation (*Horton et al., 2003*). Post-transcriptional regulation of HMGCR is mediated by sterol and nonsterol isoprenoids, which combine to accelerate the ER-associated degradation (ERAD) of HMGCR through a reaction that involves the 26S proteasome (*Nakanishi et al., 1988*; *Ravid et al., 2000*). Together, these feedback regulatory mechanisms coordinate metabolism of mevalonate to assure cells maintain constant production of nonsterol isoprenoids but avoid overaccumulation of cholesterol and other sterols.

Our group discovered that accumulation of sterols in ER membranes triggers binding of HMGCR to ER membrane proteins called Insigs (*Sever et al., 2003a*; *Sever et al., 2003b*). Subsequent ubiquitination of HMGCR by Insig-associated ubiquitin ligases (*Song et al., 2005*; *Jo et al., 2011*; *Jiang et al., 2018*) mark the enzyme for extraction across ER membranes and release into the cytosol for proteasome-mediated ERAD (*Elsabrouty et al., 2013*; *Morris et al., 2014*). GGpp augments ERAD of ubiquitinated HMGCR by enhancing its membrane extraction (*Elsabrouty et al., 2013*). Recently, we discovered that sterols also cause a subset of HMGCR molecules to bind to UBIAD1 (32). This binding protects HMGCR from accelerated ERAD, permitting continued synthesis of nonsterol isoprenoids even when cellular sterols are abundant (*Schumacher et al., 2018*). GGpp triggers release of UBIAD1 from HMGCR, which allows for maximal ERAD of HMGCR and ER-to-Golgi transport of UBIAD1 (34). Eliminating expression of UBIAD1 relieves the GGpp requirement for HMGCR ERAD, indicating the reaction is inhibited by UBIAD1. Further characterization revealed that despite its steady-state Golgi localization in GGpp-replete cells, UBIAD1 continuously cycles between the ER and Golgi. Upon sensing depletion of GGpp in membranes of the ER, UBIAD1 becomes trapped in the organelle and inhibits ERAD of HMGCR to stimulate synthesis of mevalonate for replenishment of GGpp. The physiologic relevance of UBIAD1-mediated sensing of GGpp is highlighted by the observation that the reaction appears to be disrupted in SCD. SCD-associated UBIAD1 resists GGpp-induced release from HMGCR and becomes sequestered in the ER of GGpp-replete cells (*Schumacher et al., 2015*; *Schumacher et al., 2016*). The ensuing inhibition of HMGCR ERAD, which occurs in a dominant-negative fashion, leads to a marked increase in synthesis and intracellular accumulation of cholesterol (*Schumacher et al., 2018*).

The current studies were designed to confirm the role of UBIAD1 in regulation of HMGCR ERAD and cholesterol metabolism in living animals. For this purpose, we generated mice that harbor a knockin mutation that changes asparagine-100 (N100) in UBIAD1 to a serine residue (N100S) (see *Figure 1A*). The N100S mutation in mouse UBIAD1 corresponds to the SCD-associated UBIAD1 (N102S) mutation in the human enzyme. We show here that knockin mice homozygous for the N100S mutation (designated as *Ubiad1$^{Ki/Ki}$* mice) accumulate HMGCR protein in several tissues, despite a reduction in the amount of *Hmgcr* mRNA owing to sterol accumulation and reduced activation of SREBPs. The accumulation of HMGCR protein resulted from sequestration of UBIAD1 (N100S) in the ER and inhibition of HMGCR ERAD at a post-ubiquitination step of the reaction. Aged *Ubiad1$^{Ki/Ki}$* mice exhibited signs of opacification of the cornea, which was accompanied by hallmarks of sterol overaccumulation in the tissue. These findings not only indicate that UBIAD1 modulates ERAD of HMGCR in mice through similar mechanisms previously established in cultured cells, but they also establish *Ubiad1$^{Ki/Ki}$* mice as a model for human SCD.

## Results

*Ubiad1$^{WT/Ki}$* heterozygous male and female mice (C57BL/6 $\times$ 129 genetic background) were crossed to obtain wild type (WT) and *Ubiad1$^{Ki/Ki}$* littermates. Mice homozygous for the N100S knockin mutation were born at expected Mendelian ratios. WT and *Ubiad1$^{Ki/Ki}$* littermates were externally indistinguishable and had similar body and liver weights (data not shown). Immunoblot analysis revealed that livers of male *Ubiad1$^{WT/Ki}$* and *Ubiad1$^{Ki/Ki}$* mice consuming chow diet *ad libitum* exhibited a noticeable increase (1.8- and 5.2-fold, respectively) in the amount of HMGCR protein compared to that in WT littermates (*Figure 1B*, lanes 1–3). However, the amount of *Hmgcr* mRNA was reduced approximately 40% in knockin mice (*Figure 1—figure supplement 1A*). UBIAD1 (N100S) protein also accumulated in livers of heterozygous and homozygous *Ubiad1* knockin mice (*Figure 1B*, lanes 1–3); however, this was not accompanied by an increase in hepatic *Ubiad1* mRNA (*Figure 1—figure supplement 1A*). Levels of nuclear SREBP-1 (*Figure 1B*, lanes 4–6) and SREBP-2 (lanes 7–9) were reduced in livers of *Ubiad1$^{WT/Ki}$* and *Ubiad1$^{Ki/Ki}$* mice, which coincided with reduced expression of mRNAs encoding SREBP target genes (*Figure 1—figure supplement 1A*). Cholesterol was slightly, but significantly increased in *Ubiad1$^{Ki/Ki}$* livers; however, plasma cholesterol, triglycerides, and non-esterified fatty acids as well as liver triglycerides were not significantly changed between the groups of animals (*Figure 1—figure supplement 1B*). Similar results were observed in the analysis of female *Ubiad1$^{Ki/Ki}$* mice (data not shown).

To ensure phenotypes associated with the N100S knockin mutation were not influenced by mixed genetic background, we backcrossed BL6/129 *Ubiad1$^{Ki/Ki}$* mice to C57BL/6J mice for at least six generations. For experiments described hereafter, *Ubiad1$^{WT/Ki}$* heterozygous female and male mice on the BL6 background were crossed to obtain WT and *Ubiad1$^{Ki/Ki}$* littermates. The results shown in *Figure 2A* reveal that male *Ubiad1$^{WT/Ki}$* and *Ubiad1$^{Ki/Ki}$* mice on the BL6 background accumulated hepatic HMGCR and UBIAD1 proteins (lanes 1–3), whereas levels of nuclear SREBP-1 and SREBP-2 were either unchanged (nuclear SREBP-1, lanes 4–6) or reduced (nuclear SREBP-2, lanes 7–9). HMGCR and UBIAD1 proteins accumulated and *Hmgcr* mRNA was down-regulated to varying degrees in other tissues of the knockin mice (*Figure 2B and C*). HMGCR and UBIAD1 protein accumulated to a similar extent in livers and eyes of female C57BL/6J *Ubiad1$^{Ki/Ki}$* mice (*Figure 2—figure supplement 1*).

*Table 1* shows that WT and *Ubiad1$^{Ki/Ki}$* mice had similar body and liver weights. Plasma levels of triglycerides, cholesterol, and non-esterified fatty acids (NEFAs) were slightly reduced in *Ubiad1$^{Ki/Ki}$* mice; however, these reductions were not significant. The knockin mice exhibited a small but significant increase in the amount of cholesterol in the liver (*Table 1*). We next used liquid chromatography-tandem mass spectrometry (LC-MS/MS) to measure levels of MK-4 and other nonsterol isoprenoids in livers of WT and *Ubiad1$^{Ki/Ki}$* mice (*Figure 3*). The results show that levels of MK-4 were reduced 50% relative to those observed in WT animals, despite a 4.2-fold increase in the amount of hepatic UBIAD1 protein. When normalized to the amount of UBIAD1 protein, we estimate the relative level of MK-4 was reduced by more than 80% in livers of *Ubiad1$^{Ki/Ki}$* mice. In contrast to results with MK-4, levels of geranylgeraniol (GGOH; derived from GGpp) and ubiquinone-10, were significantly increased in livers of *Ubiad1$^{Ki/Ki}$* mice. This was accompanied by a small, but significant decrease in plant-derived phylloquinone (vitamin K$_1$); bacterial-derived vitamin K$_2$ subtype

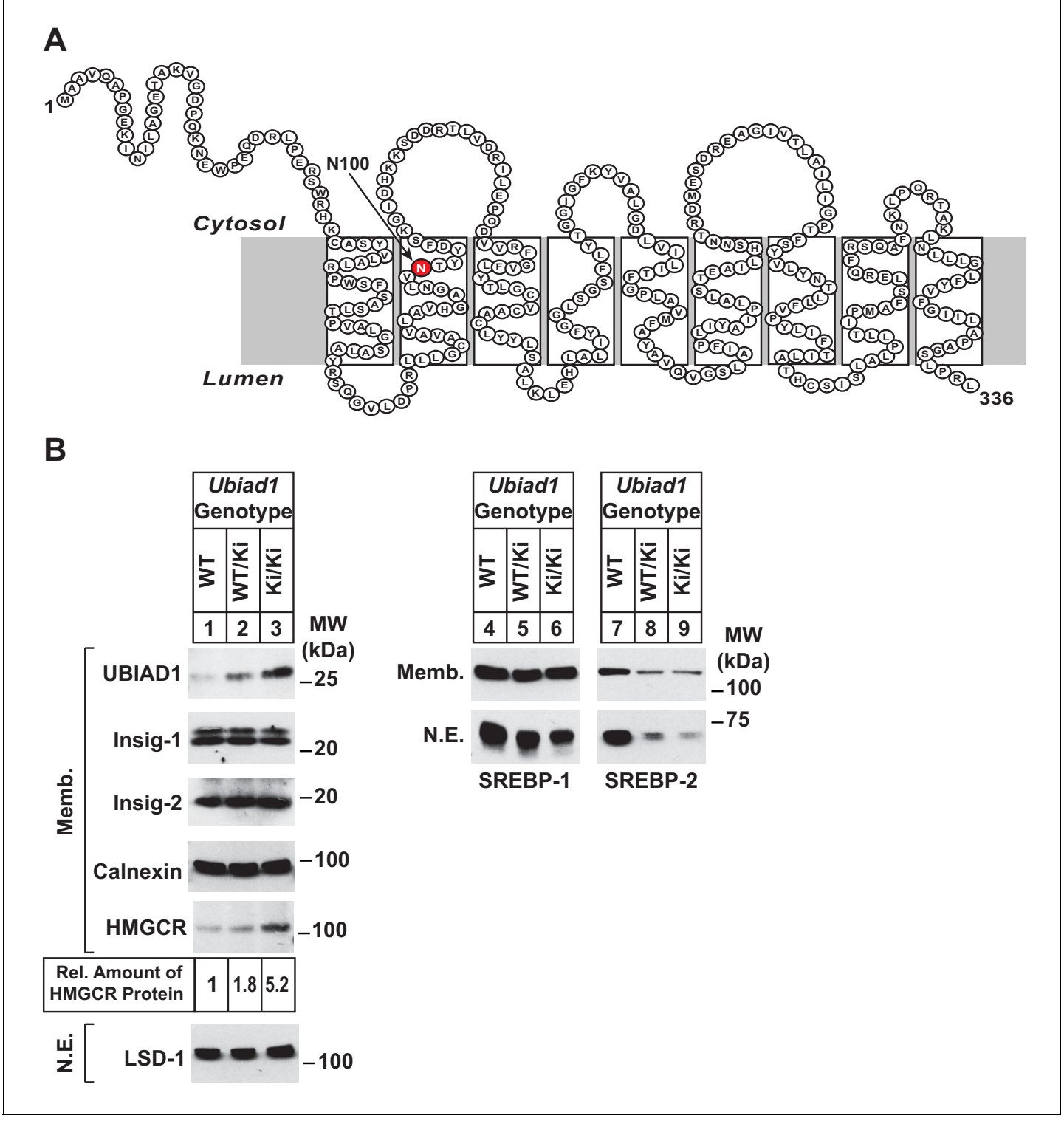

**Figure 1.** Accumulation of HMGCR protein in livers of *Ubiad1*[Ki/Ki] mice with mixed C57BL/6 × 129 genetic background. (**A**) Amino acid sequence and predicted topology of mouse UBIAD1 protein. Asparagine-100 (N100), which corresponds to the most frequently mutated amino acid residue in SCD, is enlarged, shaded in red and indicated by an arrow. (**B**) Male WT, *Ubiad1*[WT/Ki], and *Ubiad1*[Ki/Ki] littermates (8–9 weeks of age, eight mice/group) were fed an *ad libitum* chow diet prior to sacrifice. Livers of the mice were harvested and subjected to subcellular fractionation as described in 'Materials and methods.' Aliquots of resulting membrane (Memb.) and nuclear extract (N.E.) fractions (80–160 µg of total protein/lane) for each group were pooled and subjected to SDS-PAGE, followed by immunoblot analysis using antibodies against endogenous HMGCR, SREBP-1, SREBP-2, UBIAD1, Insig-1,

*Figure 1 continued on next page*

*Figure 1 continued*

Insig-2, calnexin, and LSD-1. Although shown in a separate panel, LSD-1 serves as a loading control for the nuclear SREBP immunoblots. The amount of hepatic HMGCR protein in *Ubiad1*[Ki/Ki] mice was determined by quantifying the band corresponding to HMGCR using ImageJ software.

DOI: https://doi.org/10.7554/eLife.44396.002

The following figure supplement is available for figure 1:

**Figure supplement 1.** Relative amounts of hepatic mRNAs encoding components of the Scap-SREBP pathway and lipid analysis in WT and *Ubiad1*[Ki/Ki] mice.

DOI: https://doi.org/10.7554/eLife.44396.003

menaquinone-7 (MK-7) remained unchanged in the knockin livers. MK-4 was reduced, whereas levels of GGOH and/or ubiquinone-10 were increased in kidneys, brains, spleens, and testes of *Ubiad1*[Ki/Ki] mice (*Figure 3—figure supplement 1*).

To directly study the regulation of HMGCR in the presence of UBIAD1 (N100S), we established lines of mouse embryonic fibroblasts (MEFs) from WT and *Ubiad1*[Ki/Ki] littermates. A low level of HMGCR and UBIAD1 protein was detected in membrane fractions isolated from WT MEFs cultured in sterol and nonsterol isoprenoid-replete medium containing fetal calf serum (*Figure 4A*, lane 1). Both proteins markedly accumulated in MEFs derived from *Ubiad1*[Ki/Ki] mice (lane 2). In contrast, levels of nuclear SREBP-1 and SREBP-2 were reduced in *Ubiad1*[Ki/Ki] MEFs (*Figure 4A*, compare lanes 3 and 4), which is consistent with reduced levels of *Hmgcr* mRNA and increased levels of intracellular cholesterol (*Figure 4B*). We next compared sterol-accelerated ERAD of HMGCR in *Ubiad1*[Ki/Ki] MEFs to that in MEFs derived from *Hmgcr*[Ki/Ki] mice, which harbor knockin mutations in HMGCR that prevent its sterol-induced ubiquitination (*Hwang et al., 2016*). Cells were first depleted of isoprenoids through incubation in medium containing lipoprotein-deficient serum and the HMGCR inhibitor compactin to enhance expression of HMGCR. The cells were subsequently treated in the absence or presence of the oxysterol 25-hydroxycholesterol (25-HC) prior to harvest, subcellular fractionation, and immunoblot analysis. The results show that 25-HC caused the disappearance of HMGCR from membranes of WT MEFs as expected (*Figure 4C*, lanes 1 and 2; 5 and 6). However, this disappearance was blunted in membranes from either *Ubiad1*[Ki/Ki] or *Hmgcr*[Ki/Ki] MEFs (lanes 3 and 4; 7 and 8). Despite resistance of HMGCR to sterol-accelerated ERAD, the experiment of *Figure 4D* shows that sterols continued to stimulate HMGCR ubiquitination in *Ubiad1*[Ki/Ki] MEFs. Isoprenoid-depleted cells were treated with the proteasome inhibitor MG-132 (to block degradation of ubiquitinated HMGCR) in the absence or presence of 25-HC. Cells were then harvested for preparation of detergent lysates that were immunoprecipitated with polyclonal anti-HMGCR. 25-HC caused HMGCR to become ubiquitinated in WT and *Ubiad1*[Ki/Ki] MEFs, as indicated by smears of reactivity in anti-ubiquitin immunoblots of the HMGCR immunoprecipitates (*Figure 4D*, lanes 1–6). As expected, HMGCR resisted 25-HC-induced ubiquitination in MEFs derived from *Hmgcr*[Ki/Ki] mice (compare lanes 7 and 8 with lanes 9–12).

*Figure 5A* compares expression of HMGCR in WT and *Ubiad1*[Ki/Ki] mice fed a chow diet supplemented with 1% cholesterol. The results show that cholesterol feeding led to reduced expression of HMGCR protein in membranes isolated from livers of WT mice (*Figure 5A*, lane 2); however, a significant amount of HMGCR protein remained in hepatic membranes of cholesterol-fed *Ubiad1*[Ki/Ki] mice (lane 4). The feeding regimen reduced the amount of Insig-1 protein (lanes 2 and 4) and nuclear SREBP-2 (lanes 6 and 8) in both WT and *Ubiad1*[Ki/Ki] livers. Dietary cholesterol also reduced mRNAs encoding HMGCR and other SREBP targets in livers of WT and *Ubiad1*[Ki/Ki] mice (*Figure 5—figure supplement 1*). The membrane-bound precursor and nuclear forms of SREBP-1 were induced by cholesterol feeding in livers of both lines of mice (lanes 6 and 8). This induction can be attributed to sterol-mediated activation of liver x receptors (LXRs) that modulate expression of SREBP-1c, the major SREBP-1 isoform in the mouse liver (*Repa et al., 2000*; *Liang et al., 2002*). The mRNAs encoding SREBP-1c and two other LXR targets, ABCG5 and ABCG8, were enhanced in WT and *Ubiad1*[Ki/Ki] mice fed cholesterol (*Figure 5—figure supplement 1*). In contrast to results in the liver, cholesterol-feeding failed to down-regulate levels of HMGCR protein in eyes of *Ubiad1*[Ki/Ki] mice (*Figure 5B*, lanes 3 and 4); mRNAs encoding both SREBPs and their targets were also unchanged in eyes of cholesterol-fed knockin animals (data not shown).

In *Figure 5C*, we compared cholesterol-mediated regulation of hepatic HMGCR in *Ubiad1*[Ki/Ki] and *Hmgcr*[Ki/Ki] mice. As little as 0.1% cholesterol caused a significant decrease in the amount of

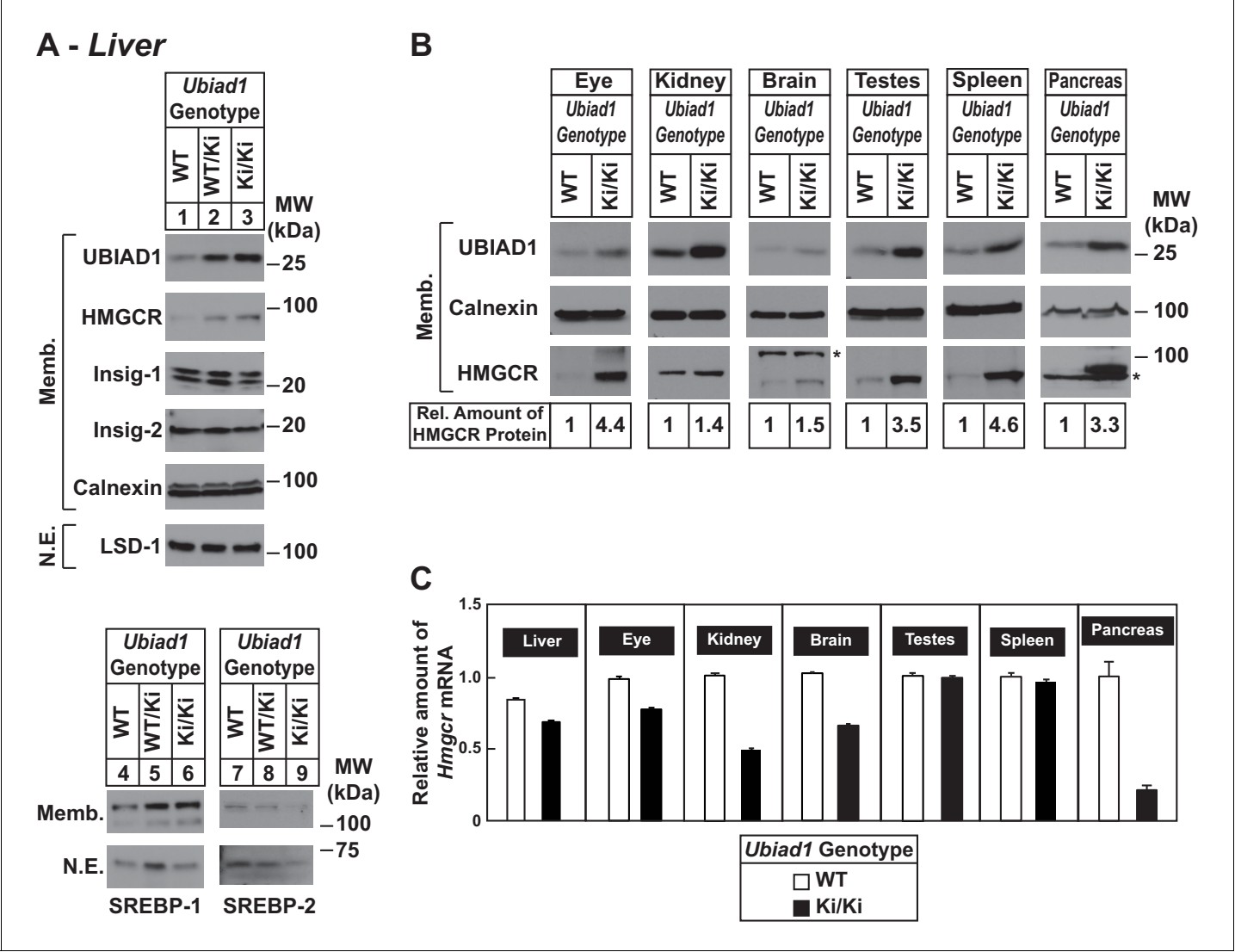

**Figure 2.** Accumulation of HMGCR protein in tissues of WT and *Ubiad1*^*Ki/Ki*^ mice with C57BL/6 genetic background. (**A and B**) Eight to nine-week old male WT, *Ubiad1*^*WT/Ki*^, and *Ubiad1*^*Ki/Ki*^ littermates (six mice/group) were fed an *ad libitum* chow diet prior to study. Aliquots of membrane (Memb.) and nuclear extract (N.E.) fractions from homogenized livers, enucleated eyes, kidneys, brains, testes, and spleens (23–50 μg of total protein/lane) were analyzed by immunoblot using antibodies against the indicated proteins. The asterisk indicates a non-specific cross-reactive band observed in the anti-HMGCR immunoblot from brain and pancreas. Although shown in separate panels, LSD-1 serves as a loading control for the nuclear SREBP-1 and SREBP-2 immunoblots. In (**B**), the amount of HMGCR protein in the indicated tissues from *Ubiad1*^*Ki/Ki*^ mice was determined by quantifying the band corresponding to HMGCR using Image J software. (**C**) For mRNA analysis, equal amounts of RNA from the indicated tissue of individual mice were subjected to quantitative real-time RT-PCR using primers against the *Hmgcr* mRNA and cyclophilin mRNA as an invariant control. *Error bars*, S.E.
DOI: https://doi.org/10.7554/eLife.44396.004

The following figure supplement is available for figure 2:

**Figure supplement 1.** Accumulation of HMGCR protein in eyes and livers of WT and *Ubiad1*^*Ki/Ki*^ mice.
DOI: https://doi.org/10.7554/eLife.44396.005

HMGCR in hepatic membranes from WT controls (*Figure 5C*, lanes b and j). HMGCR was partially resistant to 0.1% cholesterol in livers of *Hmgcr*^*Ki/Ki*^ mice (compare lanes e and f); however, higher concentrations of cholesterol (0.3% and 1%) caused complete disappearance of HMGCR from membranes (lanes g and h). The resistance of HMGCR to cholesterol feeding was more pronounced in *Ubiad1*^*Ki/Ki*^ mice; levels of the protein persisted when the animals were fed 0.1–1% cholesterol (*Figure 5C*, compare lane m with lanes n-p). Importantly, cholesterol-feeding continued to suppress levels of nuclear SREBP-2 in livers of *Hmgcr*^*Ki/Ki*^ and *Ubiad1*^*Ki/Ki*^ mice as well their WT littermates

Table 1. Comparison of wild type (WT) and *ubiad1*$^{ki/ki}$ mice.

Male *WT* and *Ubiad1*$^{Ki/Ki}$ littermates (8–9 weeks of age, eight mice/group) were fed an *ad libitum* chow diet prior to study. *WT* mice were littermates of *Ubiad1*$^{Ki/Ki}$ mice. Each value represents the mean ±S.E. of 8 values. The *p* value was calculated using Student's *t* test: *, p≤0.05.

| Parameter | WT | Ubiad1$^{Ki/Ki}$ |
| --- | --- | --- |
| Body Weight (g) | 19.8 ± 0.4 | 20.1 ± 0.6 |
| Liver Weight (g) | 1.0 ± 0.05 | 0.9 ± 0.03 |
| Plasma Triglycerides (mg/dL) | 123.6 ± 31.2 | 94.5 ± 5.7 |
| Plasma Cholesterol (mg/dL) | 100.4 ± 8.4 | 90.3 ± 9.0 |
| Plasma Nonesterified Fatty Acids (mEq/L) | 1.3 ± 0.2 | 1.1 ± 0.03 |
| Liver Triglycerides (mg/g) | 9.61 ± 1.8 | 16.3 ± 5.0 |
| Liver Cholesterol (mg/g) | 1.17 ± 0.06 | 1.65 ± 0.24* |

DOI: https://doi.org/10.7554/eLife.44396.006

(compare lanes a-d with e-h and i-l with m-p). The mRNAs encoding SREBP-2 target genes were also reduced in livers of the cholesterol-fed mice (*Figure 5D*).

We next evaluated the effect of cholesterol depletion on HMGCR levels in WT and *Ubiad1*$^{Ki/Ki}$ mice. Cholesterol depletion using lovastatin, a competitive inhibitor of HMGCR, led to the dose-dependent accumulation of HMGCR protein in livers of WT animals (*Figure 6A*, lanes a-d). Lovastatin also caused HMGCR to accumulate in livers of *Ubiad1*$^{Ki/Ki}$ mice (lanes e-h) (see *Figure 6B* for quantification). The precursor and nuclear forms of SREBP-2 were induced by lovastatin, whereas those of SREBP-1 were reduced by the treatment in both *WT* and *Ubiad1*$^{Ki/Ki}$ mice (lanes i-p). The mRNAs for SREBP-2 and its target genes (including HMGCR) were elevated in livers of lovastatin-treated animals; mRNA for SREBP-1c was reduced by the inhibitor (*Figure 6—figure supplement 1*). HMGCR protein was also increased in the eyes of lovastatin-treated *WT* mice; however, this increase required the highest concentration (0.2%) of the drug (*Figure 6C*, lanes a-d).

In the experiment of *Figure 6D*, we analyzed the subcellular localization of UBIAD1 in WT and *Ubiad1*$^{Ki/Ki}$ mice using a fractionation scheme previously utilized to isolate ER membranes from Chinese hamster ovary-K1 cells (*Radhakrishnan et al., 2008*). Liver homogenates (lysates) were first subjected to centrifugation at 3,000 X g to eliminate unbroken cells and nuclei. The resulting post-nuclear supernatants (PNS) were then applied to discontinuous sucrose gradients and centrifuged at 100,000 X g, generating two distinct membrane layers: a light membrane fraction enriched in Golgi and a heavy membrane fraction enriched in ER. Immunoblot analysis revealed the presence of UBIAD1 and the Golgi membrane protein GM-130 in the light, Golgi-enriched membrane fraction obtained from livers of WT mice fed a chow diet (*Figure 6D*, lane 4). ER-localized calnexin was observed in the ER-enriched fraction as expected (lane 5). When the mice were fed the chow diet supplemented with lovastatin (0.2%), we observed a shift in the localization of UBIAD1 from the Golgi-enriched fraction to the ER (*Figure 6D*, compare lanes 9 and 10). GM-130 remained in the Golgi-enriched fraction (lane 9) and calnexin continued to localize to the ER (lane 10) of livers from lovastatin-treated mice. In contrast to results with WT mice, UBIAD1 was concentrated in ER-enriched hepatic membranes of chow-fed *Ubiad1*$^{Ki/Ki}$ mice (*Figure 6D*, compare lanes 4 and 5 with lanes 14 and 15). The ER localization of UBIAD1 did not change when the knockin mice were challenged with lovastatin (lanes 19 and 20). Importantly, calnexin and GM-130 were localized to ER- and Golgi-enriched membranes, respectively, regardless of feeding regimen (compare lanes 14 and 15 with lanes 19 and 20).

Stereomicroscopic examinations revealed that 8–12 week-old *Ubiad1*$^{Ki/Ki}$ mice similar to those analyzed in *Figures 2*, *3*, *5* and *6* failed to exhibit significant corneal opacification that characterizes human SCD (data not shown). However, 46% (11/24) of the knockin mice exhibited signs of corneal opacification at 50 weeks of age (*Figure 7—figure supplement 1*). One of these animals manifested signs of bilateral opacification of the cornea (*Figure 7A*). None of the aged WT mice developed corneal opacification; heterozygous knockin mice were not examined (data not show**n**). Immunohisto-chemical staining with anti-HMGCR revealed a marked increase in the amount of HMGCR protein in corneas of *Ubiad1*$^{Ki/Ki}$ mice compared to their *WT* littermates (*Figure 7B*). The accumulation of

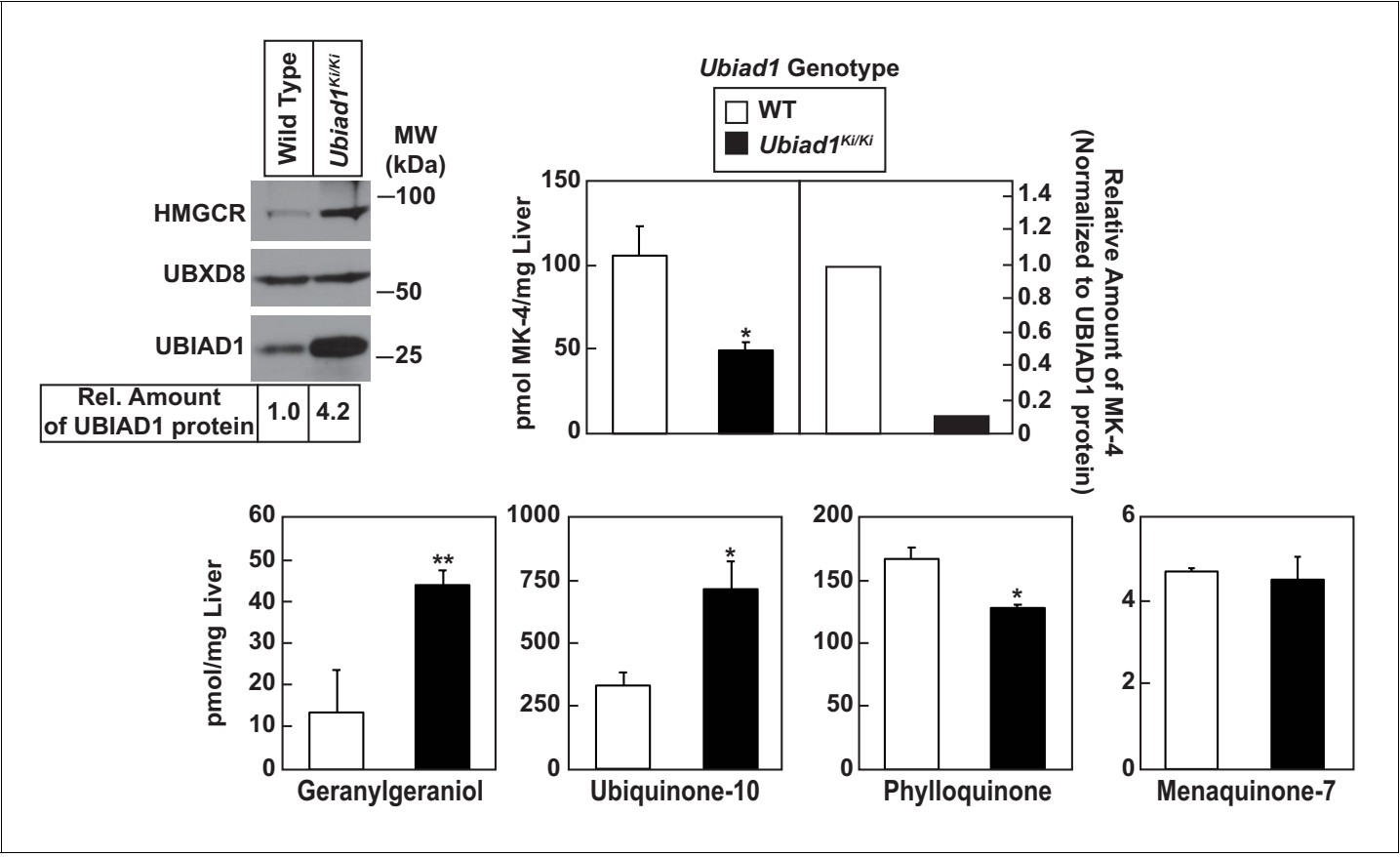

**Figure 3.** Analysis of nonsterol isoprenoids in WT and *Ubiad1*^Ki/Ki^ mice. Male mice (10–12 weeks of age, five mice/group) were fed *ad libitum* a chow diet prior to study. Livers were collected for subcellular fractionation and immunoblot analysis of resulting membrane fractions (80 µg total protein/lane) using antibodies against the indicated proteins or to determine the amount of menaquinone-4 (MK-4), geranylgeraniol, ubiquinone-10, phylloquinone, and menaquinone-7 (MK-7) by LC-MS/MS as described in 'Materials and methods.' The relative amount of hepatic MK-4 in *Ubiad1*^Ki/Ki^ mice was determined by normalizing the amount of the vitamin $K_2$ subtype to the amount of UBIAD1 protein, which was quantified using ImageJ software. *Error bars*, S.E. The *p* value was calculated using Student's *t* test: *, $p < 0.05$; **, $p < 0.01$.
DOI: https://doi.org/10.7554/eLife.44396.007
The following figure supplement is available for figure 3:

**Figure supplement 1.** Analysis of nonsterol isoprenoids in various tissues of WT and *Ubiad1*^Ki/Ki^ mice.
DOI: https://doi.org/10.7554/eLife.44396.008

HMGCR protein in *Ubiad1*^Ki/Ki^ corneas was accompanied by reduced levels of mRNAs encoding SREBP-2, HMGCR, and other cholesterol biosynthetic enzymes (*Figure 7C*). In contrast, expression of mRNAs encoding the LXR targets ABCG5, ABCG8, and ABCA1 was enhanced in *Ubiad1*^Ki/Ki^ corneas. Although the amount of total cholesterol remained unchanged in corneas of WT and *Ubiad1*^Ki/Ki^ mice, we measured a small, but significant increase in free cholesterol in the knockin mice (*Figure 7D*). Moreover, significant increases in the amount of several sterol intermediates of cholesterol synthesis including lanosterol, follicular fluid meiosis-activating sterol (FFMAS), 7- and 8-dehydrocholesterol, desmosterol, and 7-dehydrodesmosterol were observed in corneas from *Ubiad1*^Ki/Ki^ mice (*Figure 7E*).

## Discussion

The current studies provide evidence that inhibition of HMGCR ERAD directly contributes to corneal sterol accumulation and opacification that characterizes the human eye disease SCD. This conclusion was drawn from the analysis of *Ubiad1*^Ki/Ki^ mice harboring a knockin mutation (N100S) that corresponds to the SCD-associated N102S mutation in human UBIAD1 (*Figure 1A*). Consistent with our

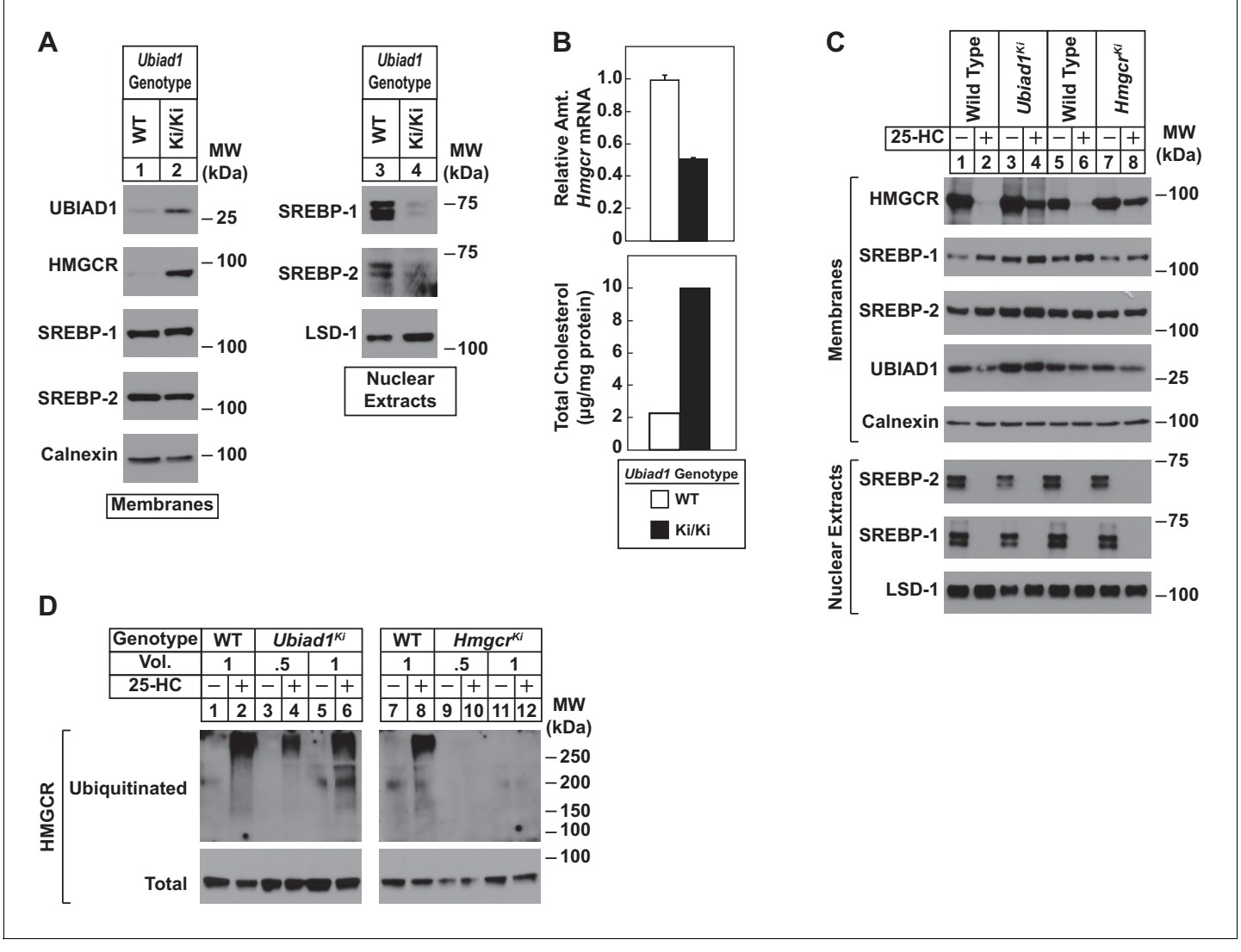

**Figure 4.** Sterol-mediated regulation of HMGCR in mouse embryonic fibroblasts (MEFs) from WT and *Ubiad1Ki/Ki* mice. MEFs from WT and *Ubiad1Ki/Ki* mice were set up for experiments on day 0 at $2 \times 10^5$ cells per 10 cm dish in MEF medium supplemented with 10% fetal calf serum (FCS). (A) On day 3, cells were harvested for subcellular fractionation. Aliquots of resulting membrane and nuclear extract fractions (35–50 µg total protein/lane) were subjected to SDS-PAGE, followed by immunoblot analysis using antibodies against the indicated proteins. (B) On day 3, cells were harvested for measurement of *Hmgcr* mRNA levels by quantitative RT-PCR and total cholesterol levels using a colorimetric assay as described in 'Materials and methods.' (C and D) On day 2, cells were depleted of isoprenoids through incubation for 16 hr at 37°C in MEF medium containing 10% lipoprotein-deficient serum, 10 µM sodium compactin, and 50 µM sodium mevalonate. The cells were subsequently treated with 1 µg/ml 25-HC as indicated; in (D), the cells also received 10 µM MG-132. (C) After 4 hr at 37°C, cells were harvested for preparation of membrane and nuclear extract fractions (35–50 µg total protein/lane) that were analyzed by immunoblot with antibodies against the indicated protein. (D) Following incubation for 1 hr at 37°C, cells were harvested, lysed in detergent-containing buffer, and immunoprecipitated with 30 µg polyclonal anti-HMGCR antibodies. Immunoprecipitated material was subjected to SDS-PAGE and immunoblot analysis with IgG-A9 (against HMGCR) and IgG-P4D1 (against ubiquitin).

DOI: https://doi.org/10.7554/eLife.44396.009

studies of human UBIAD1 (N102S) in cultured cells (*Schumacher et al., 2015*; *Schumacher et al., 2018*), mouse UBIAD1 (N100S) inhibited ERAD of HMGCR *in vivo* as indicated by accumulation of the protein in livers and other tissues of *Ubiad1Ki/Ki* mice (*Figures 1B*, *2* and *3*, and *Figure 2—figure supplement 1*). These increases in HMGCR protein occurred despite reduced levels of its mRNA (*Figures 1* and *2*), which was attributable to reduced proteolytic activation of SREBP-2 (*Figures 1B* and *2A*) resulting from accumulation of hepatic cholesterol (*Figure 1—figure supplement 1B* and *Table 1*). Corneas from 8 to 12 week-old *Ubiad1Ki/Ki* mice failed to exhibit opacification of the

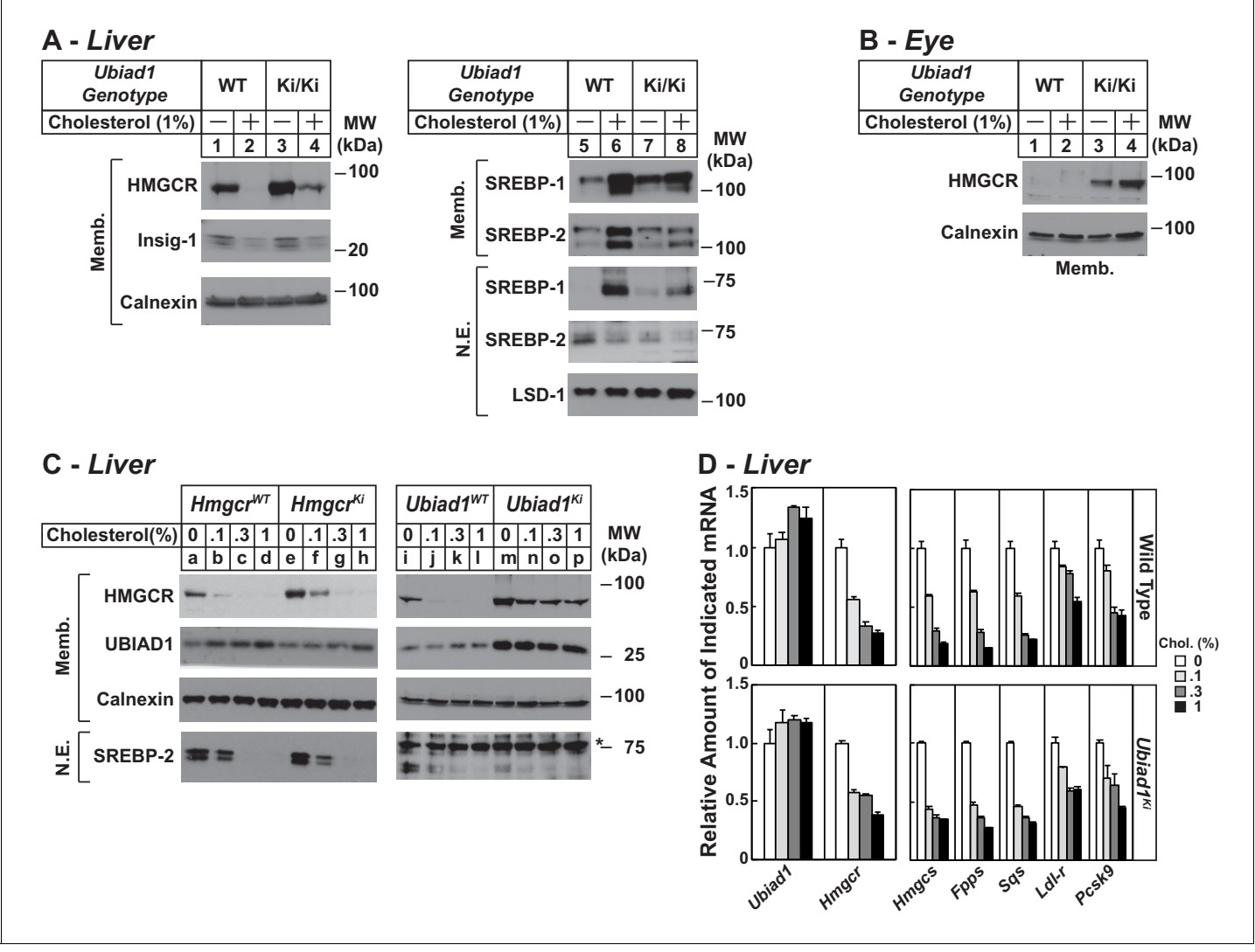

**Figure 5.** Regulation of HMGCR in livers of cholesterol-fed WT, *Ubiad1*[Ki/Ki], and *Hmgcr*[Ki/Ki] mice. Male mice (12–13 weeks of age, five mice/group) were fed an *ad libitum* chow diet supplemented with the indicated amount of cholesterol for 5 days. Aliquots of membrane (Memb.) and nuclear extract (N.E.) fractions from homogenized livers (**A and C**) or enucleated eyes (**B**) (70 µg protein/lane) were analyzed by immunoblot analysis with antibodies against the indicated proteins as described in the legend to *Figure 1*. The asterisk denotes a nonspecific band observed in the nuclear SREBP-2 immunoblot. (**D**) For mRNA analysis, equal amounts of RNA from livers of mice were subjected to quantitative real-time RT-PCR using primers against the indicated mRNAs and cyclophilin mRNA as an invariant control. *Error bars*, S.E. *Pcsk9*, proprotein convertase subtilisin/kexin type 9.

DOI: https://doi.org/10.7554/eLife.44396.010

The following figure supplement is available for figure 5:

**Figure supplement 1.** Effect of dietary cholesterol on expression of mRNAs encoding components of the Scap-SREBP pathway in livers of WT and *Ubiad1* knock-in mice.

DOI: https://doi.org/10.7554/eLife.44396.011

cornea (data not shown), despite the marked accumulation of HMGCR protein in the eye (*Figure 2B and C* and *Figure 2—figure supplement 1*). Conversely, corneal opacification was observed in aged *Ubiad1*[Ki/Ki] mice (50 weeks) (*Figure 7A*) that was accompanied by a buildup of HMGCR protein in the tissue (*Figure 7B*). This opacification occurred in the absence of an increase in total cholesterol (*Figure 7D and E*). However, *Ubiad1*[Ki/Ki] corneas displayed other hallmarks of sterol overaccumulation including an increase in levels of free cholesterol and sterol intermediates of cholesterol biosynthesis (*Figure 7D and E*), reduced levels of mRNAs for HMGCR and other cholesterol biosynthetic enzymes, and enhanced expression of mRNAs encoding ABCG5, ABCG8, and ABCA1 (*Figure 7C*).

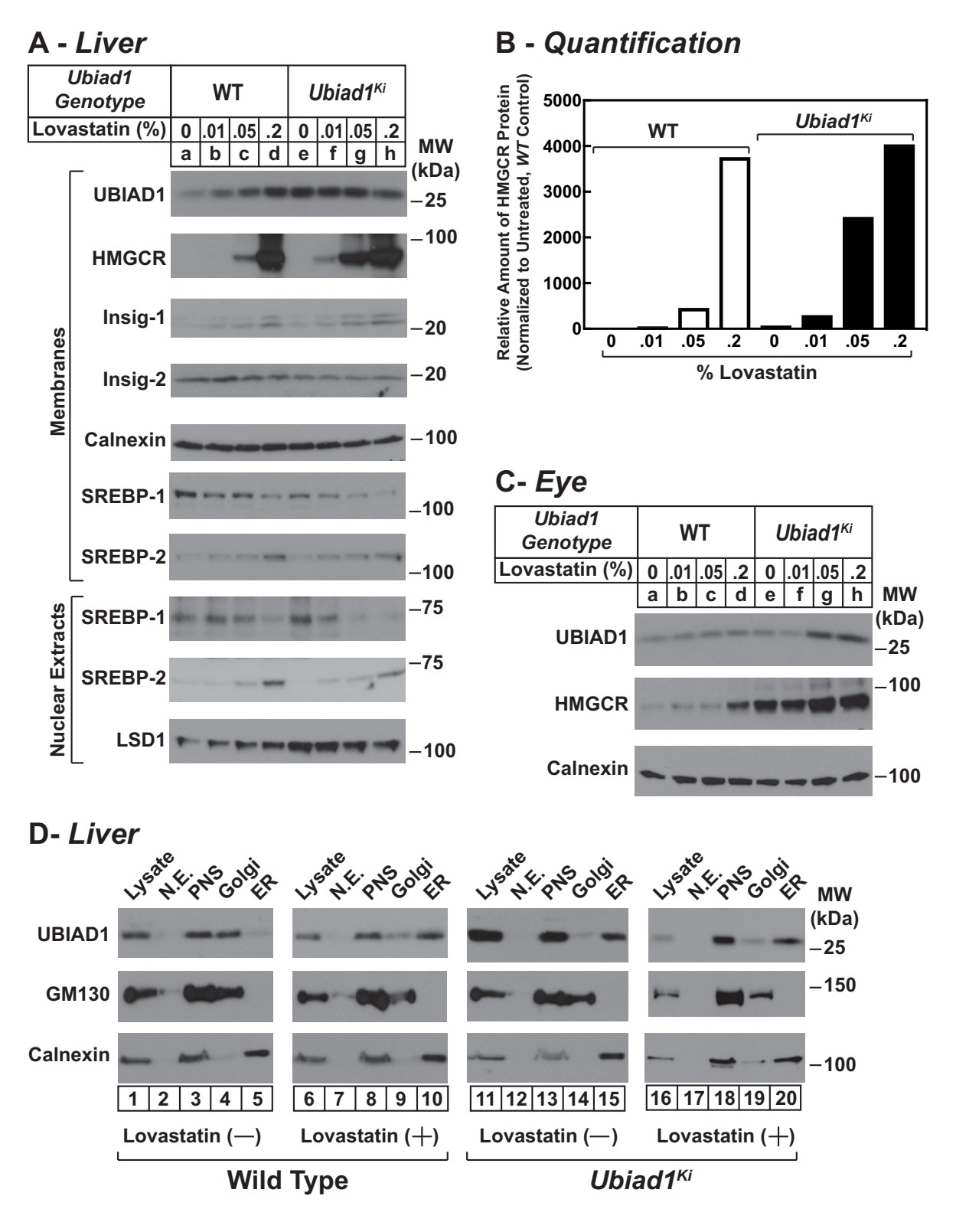

**Figure 6.** Statin-mediated regulation of HMGCR and UBIAD1 in WT and *Ubiad1^{Ki/Ki}* mice. Male mice (6–8 weeks of age, five mice/group) were fed an *ad libitum* chow diet supplemented with the indicated amount (**A and C**) or 0.2% (**D**) lovastatin for 5 days. (**A and C**) Aliquots of membrane and nuclear extract fractions from homogenized livers (**A**) or enucleated eyes (**C**) (70 µg protein/lane) were analyzed by immunoblot analysis with antibodies against the indicated proteins. In (**B**), the amount of HMGCR protein in livers of *Ubiad1^{Ki/Ki}* mice shown in (**A**) was determined by quantifying the band

*Figure 6 continued on next page*

Figure 6 continued

corresponding to HMGCR using Image J software and normalizing to the amount of the protein in untreated WT controls. (D) Post nuclear supernatants (PNS) obtained from liver homogenates were fractionated on a discontinuous sucrose gradient (7.5–45%) that yielded a light membrane fraction enriched in Golgi and a heavy membrane fraction enriched in ER. Aliquots of the homogenates (lysate), nuclear extracts (N.E.), PNS, Golgi-enriched membranes, and ER-enriched membranes were subjected to SDS-PAGE, followed by immunoblot analysis with antibodies against the indicated proteins.

DOI: https://doi.org/10.7554/eLife.44396.012

The following figure supplement is available for figure 6:

**Figure supplement 1.** Effect of lovastatin on expression of mRNAs encoding components of the Scap-SREBP pathway in livers of WT and *Ubiad1* knock-in mice.

DOI: https://doi.org/10.7554/eLife.44396.013

Based on these observations, we conclude that UBIAD1 (N100S)-mediated inhibition of HMGCR ERAD enhances flux through the cholesterol biosynthetic pathway, thereby initiating sterol accumulation and development of corneal opacification. Our current findings suggest the slow progression of corneal opacification in *Ubiad1*$^{Ki/Ki}$ mice results from reduced activation of SREBP-2, which leads to reduced expression of genes encoding cholesterol biosynthetic enzymes. Studies are now underway to monitor progression of corneal opacification in *Ubiad1*$^{Ki/Ki}$ mice > 50 weeks of age. We anticipate that as the disorder progresses, free cholesterol will continue to accumulate, initiating formation and accumulation of cholesterol esters, which will further exacerbate opacification of the cornea. Additionally, *Ubiad1*$^{Ki/Ki}$ mice will be challenged with high cholesterol/Western diets to determine whether dietary cholesterol contributes to progression of corneal opacification in SCD.

A preliminary analysis of UBIAD1 (N100S) knockin mice has been recently described (*Dong et al., 2018*). However, this study lacks the molecular characterization of HMGCR and documentation of age-dependent corneal opacification in the knockin animals. Instead, the authors report evidence for mitochondrial damage and accumulation of mitochondrial-localized glycerophosphoglycerols in knockin corneas. The authors conclude that mitochondrial dysfunction is linked to the abnormal deposition of cholesterol in corneas of SCD patients, constrasting our conclusion the response results from inhibition of HMGCR ERAD. The possibility exists that this mitochondrial dysfunction is secondary to cholesterol accumulation and/or impairment of MK-4 synthesis (see *Figure 3*) in *Ubiad1*$^{Ki/Ki}$ mice. Further investigation is required to resolve this discrepancy.

Examination of *Ubiad1*$^{Ki/Ki}$-derived MEFs cultured under isoprenoid-replete conditions yielded results remarkably similar to those obtained with whole mouse tissues – marked accumulation of HMGCR protein, overproduction of cholesterol, reduced levels of *Hmgcr* mRNA, and reduced activation of SREBPs (*Figure 4A and B*). HMGCR was also refractory to accelerated ERAD stimulated by the oxysterol 25-HC in *Ubiad1*$^{Ki/Ki}$ MEFs and *Hmgcr*$^{Ki/Ki}$ MEFs, which were derived from mice expressing ubiquitination-resistant HMGCR (*Hwang et al., 2016*) (*Figure 4C*). However, 25-HC continued to stimulate ubiquitination of HMGCR in *Ubiad1*$^{Ki/Ki}$ MEFs, but not in those derived from *Hmgcr*$^{Ki/Ki}$ mice (*Figure 4D*). In addition to providing direct evidence that SCD-associated UBIAD1 inhibits ERAD of HMGCR, these results indicate that the inhibition results from a block in a post-ubiquitination step of the reaction (*Schumacher et al., 2015*).

Our previous characterization of *Hmgcr*$^{Ki/Ki}$ mice indicated that dietary cholesterol reduces levels of HMGCR protein primarily by inhibiting activation of SREBP-2 (35); similar results were obtained in the current study (*Figure 5C*). In contrast, levels of HMGCR protein persisted in livers of *Ubiad1*$^{Ki/Ki}$ mice fed cholesterol, even though the feeding regimen continued to block proteolytic activation of SREBP-2 (*Figure 5A and C*) and reduce levels of mRNAs for HMGCR and other SREBP-2 targets (*Figure 5—figure supplement 1*). Previous studies found that HMGCR is subjected to sterol-independent ubiquitination and ERAD in cultured cells (*Doolman et al., 2004*). Thus, we speculate that in livers of *Hmgcr*$^{Ki/Ki}$ mice, ubiquitination-resistant HMGCR continues to become degraded through this sterol-independent ERAD pathway. However, both sterol-independent and sterol-dependent pathways for HMGCR ERAD are blocked in *Ubiad1*$^{Ki/Ki}$ mice, owing to the ability of UBIAD1 (N100S) to inhibit post-ubiquitination steps of the reaction (see *Figure 4D*). As a result of this inhibition, levels of HMGCR protein persist in livers of cholesterol-fed *Ubiad1*$^{Ki/Ki}$ mice.

Okano and co-workers previously attempted to generate mice lacking *Ubiad1*; however, *Ubiad1*-deficient embryos failed to survive past embryonic day 7.5 (40). *Ubiad1*$^{-/-}$ embryonic stem cells failed

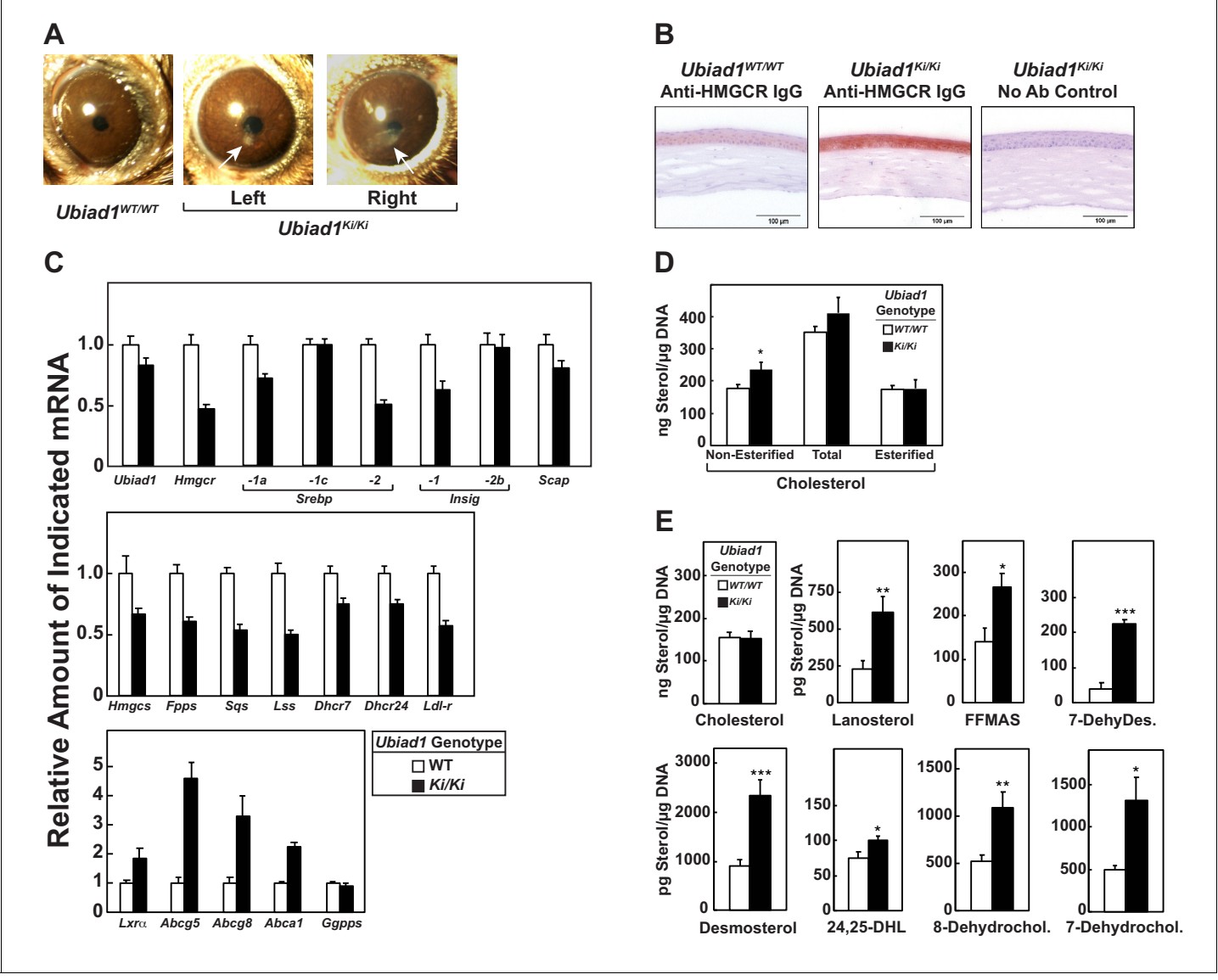

**Figure 7.** *Ubiad1^{Ki/Ki}* mice exhibit signs of corneal opacification upon aging. (A) Male and female mice (15 WT, 24 *Ubiad1^{Ki/Ki}*, 50 weeks of age) consuming an *ad libitum* chow diet were analyzed by stereomicroscopic examination. Corneal opacification is indicated by white arrows. (B–E) Mice analyzed in (A) were sacrificed, corneas were then harvested and analyzed by immunohistochemical staining with anti-HMGCR polyclonal antibodies (B), quantitative RT-PCR (C), and LC-MS/MS (D and E) as described in the legend to *Figure 1* and 'Materials and methods.' *Error bars*, S.E. The *p* value was calculated using Student's *t* test: *, p < 0.05; **, p < 0.01; ***, p 0.005. *Dhcr7*, 7-dehydrocholesterol reductase; *Dhcr24*, 24-dehydrocholesterol reductase; 7-DehyDes., 7-dehydrodesmosterol; 8-Dehydrochol., 8-dehydrocholesterol; 7-Dehydrochol., 7-dehydrocholesterol.

DOI: https://doi.org/10.7554/eLife.44396.014

The following figure supplement is available for figure 7:

**Figure supplement 1.** *Ubiad1^{Ki/Ki}* mice exhibit signs of corneal opacification upon aging.

DOI: https://doi.org/10.7554/eLife.44396.015

to synthesize MK-4, suggesting the compound plays a pivotal role in development. Taking this and the observation that human UBIAD1 (N102S) is defective in MK-4 synthetic activity (*Hirota et al., 2015*), we were somewhat surprised that *Ubiad1^{Ki/Ki}* mice were born at normal Mendelian ratios and appear grossly normal. Throughout our studies, we noticed that UBIAD1 (N100S) protein accumulated in all tissues of *Ubiad1^{Ki/Ki}* mice (see *Figures 1–3*). Similar results have been observed for SCD-associated variants of human UBIAD1 in transfected cells; stabilization of SCD-associated UBIAD1 has been attributed to ER sequestration and protection from autophagic degradation from the Golgi

(Jun, D.-J. and DeBose-Boyd, R.A., unpublished observations). The level of MK-4 was significantly reduced, but not absent, in various tissues of *Ubiad1*$^{Ki/Ki}$ mice (*Figure 3* and *Figure 3—figure supplement 1*). We speculate that despite reduced enzymatic activity, accumulated UBIAD1 (N100S) produces sufficient amounts of MK-4 to support development and survival of *Ubiad1*$^{Ki/Ki}$ mice. Accumulation of SCD-associated UBIAD1 in the ER has important implications for the pathology of the disease. SCD is an autosomal dominant disorder and our previous studies revealed that SCD-associated variants of UBIAD1 inhibit HMGCR ERAD in a dominant-negative fashion (*Schumacher et al., 2015*; *Schumacher et al., 2016*). Enhanced stability of SCD-associated UBIAD1 owing to its ER sequestration helps to explain how cholesterol accumulation occurs in corneas of SCD patients harboring heterozygous UBIAD1 mutations.

The significance of the HMGCR regulatory system is evidenced by the widespread use of statins to lower plasma LDL-cholesterol and reduce the incidence of atherosclerotic cardiovascular disease (*Stossel, 2008*). However, statins block synthesis of sterol and nonsterol isoprenoids that mediate feedback regulation of HMGCR, causing the enzyme's accumulation in livers of humans and animals (*Kita et al., 1980*; *Reihnér et al., 1990*) (*Figure 6*). This accumulation partially overcomes inhibitory effects of statins, which allows for continued synthesis of cholesterol that limits cholesterol-lowering (*Schonewille et al., 2016*; *Goldberg et al., 1990*; *Engelking et al., 2006*). We showed previously that the statin-induced increase in HMGCR was blunted 5-fold in *Hmgcr*$^{Ki/Ki}$ versus WT mice, suggesting inhibition of ERAD significantly contributes to the response (*Hwang et al., 2016*). Our current findings argue that statin-induced accumulation of HMGCR results in part, from depletion of GGpp from ER membranes. This was indicated by accumulation and ER sequestration of UBIAD1 in livers of lovastatin-treated mice, which coincided with accumulation of HMGCR (*Figure 6A and C*). UBIAD1 (N100S), which resists GGpp-induced release from HMGCR, accumulated and remained sequestered in the ER, regardless of whether *Ubiad1*$^{Ki/Ki}$ mice were treated in the absence or presence of lovastatin (*Figure 6A and C*). The resultant inhibition of HMGCR ERAD led to accumulation of the protein and overproduction of sterol and nonsterol isoprenoids in various tissues of *Ubiad1*$^{Ki/Ki}$ mice (see *Figures 3* and *7*, and *Figure 3—figure supplement 1*). Thus, GGpp-regulated, ER-to-Golgi transport of UBIAD1 regulates HMGCR ERAD to coordinate synthesis of sterol and nonsterol isoprenoids in mice through similar mechanisms previously described in cultured cells (*Schumacher et al., 2018*). The current results indicate that modulating ER-to-Golgi transport of UBIAD1 may have therapeutic value. For example, agents that mimic GGpp in stimulating ER-to-Golgi transport of UBIAD1 should relieve inhibition of ERAD and prevent accumulation of HMGCR that limits the effectiveness of statins in lowering plasma levels of LDL-cholesterol. Agents that restore Golgi localization of SCD-associated UBIAD1 or block its interaction with HMGCR may prevent or retard cholesterol accumulation and corneal opacification associated with SCD. The establishment of a mouse model for SCD described here will prove instrumental in the identification and characterization of such molecules.

# Materials and methods

**Key resources table**

| Reagent type (species) or resource | Designation | Source or reference | Identifiers | Additional information |
|---|---|---|---|---|
| Genetic reagent (*M. musculus*) | Mouse/*Ubiad1*$^{Ki/Ki}$ (UBIAD1 (N100S)): C57BL/6J | This paper | N/A | Heterozygous knockin mice harboring mutations in the endogenous Ubiad1 gene that change Asparagine-100 to a Serine residue |

*Continued on next page*

Continued

| Reagent type (species) or resource | Designation | Source or reference | Identifiers | Additional information |
|---|---|---|---|---|
| Genetic reagent (*M. musculus*) | Mouse/*Ubiad1*<sup>Ki/Ki</sup> (UBIAD1 (N100S)): C57BL/6 | This paper | | Homozygous knockin mice harboring mutations in the endogenous Ubiad1 gene that change Asparagine-100 to a Serine residue |
| Genetic reagent (*M. musculus*) | Mouse/*Hmgcr*<sup>Ki/Ki</sup> (HMGCR K89R/ K248R):C57BL/6 | PMID: 27129778 | N/A | |
| Cell line | Mouse Embryonic Fibroblast-*Ubiad1*<sup>WT/WT</sup> | This paper | N/A | Mouse embryonic fibroblasts from wild type C57BL/ 6 mice |
| Cell line | Mouse Embryonic Fibroblast-*Ubiad1*<sup>Ki/Ki</sup> | This paper | N/A | Mouse embryonic fibroblasts from *Ubiad1Ki/Ki* C57BL/6 mice |
| Cell line | Mouse Embryonic Fibroblast-*Hmgcr*<sup>WT/WT</sup> | This paper | N/A | Mouse embryonic fibroblasts from wild type C57BL/ 6 mice |
| Cell line | Mouse Embryonic Fibroblast-*Hmgcr*<sup>Ki/Ki</sup> | This paper | N/A | Mouse embryonic fibroblasts from *HmgcrKi/Ki* C57BL/6 mice |
| Antibody | Rabbit monoclonal anti-SREBP-1 | PMID: 28244871 | IgG-20B12 | |
| Antibody | Rabbit monoclonal anti-SREBP-2 | PMID: 25896350 | IgG-22D5 | |
| Antibody | Rabbit polyclonal anti-UBIAD1 | This paper | IgG-205 | Rabbit polyclonal antibody raised against amino acids 2–21 of mouse UBIAD1; used at 1–5 µg/ml for immunoblots |
| Antibody | Rabbit polyclonal anti-HMGCR | PMID: 27129778 | IgG-839c | used at 1–5 µg/ml for immunoblots |
| Antibody | Mouse monoclonal anti-HMGCR | PMID: 22143767 | IgG-A9 | used at 1–5 µg/ml for immunoblots |
| Antibody | Rabbit polyclonal anti-Insig-1 | PMID: 27129778 | anti-Insig-1 | used at 1:1000 dilution for immunoblots |
| Antibody | Rabbit polyclonal anti-Insig-2 | This paper | IgG-492 | Rabbit polyclonal antibody raised against a C-terminal peptide (CKVIPEKSHQE) of hamster Insig-2; used at 5 µg/ml for immunoblots |
| Antibody | Rabbit polyclonal anti-UBXD8 | PMID: 27129778 | IgG-819 | used at 1–5 µg/ml for immunoblots |

*Continued on next page*

*Continued*

| Reagent type (species) or resource | Designation | Source or reference | Identifiers | Additional information |
|---|---|---|---|---|
| Antibody | Rabbit polyclonal anti-Calnexin | Novus Biologicals | Cat#NB100-1965; RRID:AB_10002123 | used at 1–5 µg/ml for immunoblots |
| Antibody | Rabbit polyclonal anti-GM130 | Abcam | Cat#ab 30637; RRID:AB_732675 | used at 1–5 µg/ml for immunoblots |
| Antibody | Rabbit polyclonal anti-LSD-1 | Cell Signaling Technology | Cat#2139; RRID:AB_2070135 | used at 1–5 µg/ml for immunoblots |
| Antibody | Mouse monoclonal anti-ubiquitin (IgG-P4D1) | Santa Cruz | Cat#SC8017; RRID:AB_628423 | used at 1–5 µg/ml for immunoblots |
| Recombinant DNA reagent | | | | |
| Sequence-based reagent | *Ubiad1*$^{Ki/Ki}$ genotyping primers: Forward, GGAACACTTGGCTCTCATCT; Reverse, GGGAGCAGTGTTCATAATCC | This paper | N/A | Genotyping was determined by PCR analysis of genomic DNA prepared from tails of mice. |
| Sequence-based reagent | *Hmgcr*$^{Ki/Ki}$ genotyping primers: K89R- Forward, GTCCATGAACATGTTCACCG; Reverse, CAGCACGTCCTATTGGCAGA K248R – Forward, TCGGTGATGTTCCAGTCTTC; Reverse, GGTGGCAAACACCTTGTATC | PMID: 27129778 | N/A | |
| Sequence-based reagent (qRT-PCR) | UBIAD1 Forward, GACAGAACTTTGGTGGACAGAATTC; Reverse, CAGCCCAAGGTGTAGAGGAAGA | Integrated DNA Technologies | N/A | |
| Sequence-based reagent (qRT-PCR) | SREBP-1a Forward, GGCCGAGATGTGCGAACT; Reverse, TTGTTGATGAGCTGGAGCATGT | Integrated DNA Technologies | N/A | |
| Sequence-based reagent (qRT-PCR) | SREBP-1c Forward, GGAGCCATGGATTGCACATT; Reverse, GGCCCGGGAAGTCACTGT | Integrated DNA Technologies | N/A | |
| Sequence-based reagent (qRT-PCR) | SREBP-2 Forward, GCGTTCTGGAGACCATGGA; Reverse, ACAAAGTTGCTCTGAAAACAAATCA | Integrated DNA Technologies | N/A | |
| Sequence-based reagent (qRT-PCR) | HMGCR Forward, CTTGTGGAATGCCTTGTGATTG; Reverse, AGCCGAAGCAGCACATGAT | Integrated DNA Technologies | N/A | |
| Sequence-based reagent (qRT-PCR) | Insig-1 Forward, TCACAGTGACTGAGCTTCAGCA; Reverse, TCATCTTCATCACACCCAGGAC | Integrated DNA Technologies | N/A | |
| Sequence-based reagent (qRT-PCR) | Insig-2a Forward, CCCTCAATGAATGTACTGAAGGATT; Reverse, TGTGAAGTGAAGCAGACCAATGT | Integrated DNA Technologies | N/A | |
| Sequence-based reagent (qRT-PCR) | Insig-2b Forward, CCGGGCAGAGCTCAGGAT; Reverse, GAAGCAGACCAATGTTTCAATGG | Integrated DNA Technologies | N/A | |
| Sequence-based reagent (qRT-PCR) | SCAP Forward, ATTTGCTCACCGTGGAGATGTT; Reverse, GAAGTCATCCAGGCCACTACTAATG | Integrated DNA Technologies | N/A | |
| Sequence-based reagent (qRT-PCR) | HMGCS Forward, GCCGTGAACTGGGTCGAA; Reverse, GCATATATAGCAATGTCTCCTGCAA | Integrated DNA Technologies | N/A | |

*Continued on next page*

*Continued*

| Reagent type (species) or resource | Designation | Source or reference | Identifiers | Additional information |
|---|---|---|---|---|
| Sequence-based reagent (qRT-PCR) | FPPS Forward, ATGGAGATGGGCGAGTTCTTC; Reverse, CCGACCTTTCCCGTCACA | Integrated DNA Technologies | N/A | |
| Sequence-based reagent (qRT-PCR) | SqS Forward, CCAACTCAATGGGTCTGTTCCT; Reverse, TGGCTTAGCAAAGTCTTCCAACT | Integrated DNA Technologies | N/A | |
| Sequence-based reagent (qRT-PCR) | LDLR Forward, AGGCTGTGGGCTCCATAGG; Reverse, TGCGGTCCAGGGTCATCT | Integrated DNA Technologies | N/A | |
| Sequence-based reagent (qRT-PCR) | PCSK9 Forward, CAGGCGGCCAGTGTCTATG; Reverse, GCTCCTTGATTTTGCATTCCA | Integrated DNA Technologies | N/A | |
| Sequence-based reagent (qRT-PCR) | ACS Forward, GCTGCCGACGGGATCAG; Reverse, TCCAGACACATTGAGCATGTCAT | Integrated DNA Technologies | N/A | |
| Sequence-based reagent (qRT-PCR) | ACC1 Forward, TGGACAGACTGATCGCAGAGAAAG; Reverse, TGGAGAGCCCCACACACA | Integrated DNA Technologies | N/A | |
| Sequence-based reagent (qRT-PCR) | FAS Forward, GCTGCGGAAACTTCAGGAAAT; Reverse, AGAGACGTGTCACTCCTGGACTT | Integrated DNA Technologies | N/A | |
| Sequence-based reagent (qRT-PCR) | SCD1 Forward, CCGGAGACCCCTTAGATCGA; Reverse, TAGCCTGTAAAAGATTTCTGCAAACC | Integrated DNA Technologies | N/A | |
| Sequence-based reagent (qRT-PCR) | GPAT Forward, CAACACCATCCCCGACATC; Reverse, GTGACCTTCGATTATGCGATCA | Integrated DNA Technologies | N/A | |
| Sequence-based reagent (qRT-PCR) | LXRα Forward, TCTGGAGACGTCACGGAGGTA; Reverse, CCCGGTTGTAACTGAAGTCCTT | Integrated DNA Technologies | N/A | |
| Sequence-based reagent (qRT-PCR) | ABCG5 Forward, TGGATCCAACACCTCTATGCTAAA; Reverse, GGCAGGTTTTCTCGATGAACTG | Integrated DNA Technologies | N/A | |
| Sequence-based reagent (qRT-PCR) | ABCG8 Forward, TGCCCACCTTCCACATGTC; Reverse, ATGAAGCCGGCAGTAAGGTAGA | Integrated DNA Technologies | N/A | |
| Sequence-based reagent (qRT-PCR) | GGPS Forward, CGTCTACTTCCTTGGACTGGAAA; Reverse, AGCTGGCGTGTGAAAAGCTT | Integrated DNA Technologies | N/A | |
| Sequence-based reagent (qRT-PCR) | Cyclophilin Forward, TGGAGAGCACCAAGACAGACA; Reverse, TGCCGGAGTCGACAATGAT | Integrated DNA Technologies | N/A | |
| Commercial assay or kit | TaqMan Reverse Transcription | Applied Biosystems | Cat#N8080234 | |
| Commercial assay or kit | Power SYBR Green PCR Master Mix | Applied Biosystems | Cat#4367659 | |
| Commercial assay or kit | Cholesterol/ Cholesterol Ester Assay Kit - Quantitation | Abcam | Cat#ab65359 | |

*Continued on next page*

*Continued*

| Reagent type (species) or resource | Designation | Source or reference | Identifiers | Additional information |
|---|---|---|---|---|
| Chemical compound, drug | Cholesterol | Bio-Serv; | Cat#5180; | |
| Chemical compound, drug | | Sigma-Aldrich | Cat#C8667 | |
| Chemical compound, drug | Coenzyme Q-10 | Cerilliant | Cat#V-060 | |
| Chemical compound, drug | Geranylgeraniol | Sigma-Aldrich | Cat#G3278 | |
| Chemical compound, drug | Geranylgeranyl pyrophosphate | Cayman Chemical Company | Cat#63330 | |
| Chemical compound, drug | Lovastatin | Abblis Chemicals LLC, Houston, TX | Cat#AB1004848 | |
| Chemical compound, drug | Menaquinone-4 | Sigma-Aldrich | Cat#809896 | |
| Chemical compound, drug | | Cerilliant | Cat#V-031 | |
| Chemical compound, drug | Menaquinone-7 | Cerilliant | Cat#V-044 | |
| Chemical compound, drug | Phylloquinone (Vitamin K1) | Cerilliant | Cat#V-030 | |
| Chemical compound, drug | 25-Hydroxy cholesterol | Avanti Polar Lipids | Cat#700019P | |
| Software, algorithm | Image Studio v5.0 | LiCor Biosciences | | |
| Software, algorithm | Image J (Fiji) | NIH | | |

## Animals

*Ubiad1$^{Ki/Ki}$* mice, which harbor a nucleotide substitution in the *Ubiad1* gene that changes asparagine-100 to a serine residue (N100S), were generated by the Gene Targeting and Transgenic Facility at the Howard Hughes Medical Institute Janelia Research Campus (Ashburn, VA). *Ubiad1$^{WT/WT}$* and *Ubiad1$^{Ki/Ki}$* littermates were obtained for experiments from intercrosses of *Ubiad1$^{WT/Ki}$* male and female mice that were hybrids of C57BL/6J and 129Sv/Ev strains. *Ubiad1$^{Ki/Ki}$* mice on the mixed BL6/129 background were backcrossed to C57BL/6J mice for at least six generations. Intercrosses of *Ubiad1$^{WT/Ki}$* male and female mice on the BL6 background were conducted to obtain *Ubiad1$^{WT/WT}$* and *Ubiad1$^{Ki/Ki}$* littermates for experiments. *Hmgcr$^{Ki/Ki}$* mice harbor homozygous knockin mutations in which lysine residues 89 and 248 are replaced with arginines (*Hwang et al., 2016*). All mice were housed in colony cages with a 12 hr light/12 hr dark cycle and fed Teklad Mouse/Rat diet 2018 from Harlan Teklad (Madison, WI). Genomic DNA was extracted from tails of *Ubiad1$^{Ki/Ki}$* and *Hmgcr$^{Ki/Ki}$* mice using DNeasy Blood and Tissue kit (Qiagen, Venlo, Netherlands) according to the manufacturer's protocol. To genotype *Ubiad1$^{Ki/Ki}$* mice, genomic DNA from tails was used for PCR with the following primers: Forward, GGAACACTTGGCTCTCATCT; Reverse, GGGAGCAGTGTTCATAATCC. *Hmgcr$^{Ki/Ki}$* mice were genotyped as described previously (*Hwang et al., 2016*). The levels of plasma and liver cholesterol and triglycerides were measured by the Metabolic Phenotyping Core at the

University of Texas Southwestern Medical Center (UTSWMC) in blood drawn from the vena cava after mice were anesthetized in a bell-jar atmosphere containing isoflurane. For the cholesterol feeding studies, mice were fed a chow diet (Teklad Mouse/Rat 2018, 0% cholesterol) or chow diet supplemented with 0.1, 0.3, or 1% cholesterol for 5 days prior to study. For lovastatin feeding studies, mice were fed Teklad Mouse/Rat diet (Harlan Teklad Premier Laboratory Diets, Madison, WI) or the identical diet supplemented with 0.01, 0.05, or 0.2% lovastatin (Abblis Chemicals LLC, Houston, TX). All animal experiments were performed with the approval of the Institutional Animal Care and Use Committee at UTSWMC.

## Quantitative Real-Time PCR

Total RNA was prepared from mouse tissues using the RNA STAT-60 kit (TEL-TEST 'B', Friendswood, TX). Equal amounts of RNA from individual mice were treated with DNase I (DNA-free, Ambion/Life Technologies, Grand Island, NY). First strand cDNA was synthesized from 10 µg of DNase I-treated total RNA with random hexamer primers using TaqMan Reverse Transcription Reagents (Applied Biosystems/Roche, Branchburg, NJ). Specific primers for each gene were designed using Primer Express software (Life Technologies). The real-time RT-PCR reaction was set up in a final volume of 20 µl containing 20 ng of reverse-transcribed total RNA, 167 nM of the forward and reverse primers, and 10 µl of 2X SYBR Green PCR Master Mix (Life Technologies). The relative amount of all mRNAs was calculated using the comparative threshold cycle ($C_T$) method. Mouse cyclophilin mRNA was used as the invariant control.

## Generation of Mouse Embryonic Fibroblasts (MEFs)

The protocol for establishing MEFs from $Ubiad1^{Ki/Ki}$ and $Hmgcr^{Ki/Ki}$ mice was adapted from that described previously (*Jozefczuk et al., 2012*; *Cautivo et al., 2016*). Briefly, pregnant $Ubiad1^{+/Ki}$ and $Hmgcr^{+/Ki}$ female mice were sacrificed 13.5 days post coitum and uterine horns were harvested in cold PBS. In a tissue culture hood under aseptic conditions, the uterine horns were placed into a Petri dish and each embryo was separated from its placenta and embryonic sac. The head of embryo was removed and saved for genotyping. The remainder of the embryo was washed with PBS and minced. The minced tissues were incubated in the presence of 0.05% Trypsin/EDTA and DNase I for 30 min at 37°C with intermittent agitation. The reaction was stopped by adding MEF media containing DMEM 4.5 g/L glucose, 10% FCS, and 1% Penicillin/streptomycin (10,000 U/ml). Cells were pelleted at 300 X $g$ for 5 min and plated onto 0.2% gelatin coated dishes; these cells were designated passage 0 and frozen. All MEFs tested negative for mycoplasma contamination. Passages 2–5 were used for experiments. The level of intracellular cholesterol in $Ubiad1^{WT/WT}$ and $Ubiad1^{Ki/Ki}$ MEFs was determined using Cholesterol/Cholesterol Ester Assay Kit (Abcam).

## Subcellular Fractionation, Immunoblot Analysis, and Immunohistochemistry

Approximately 50 mg of frozen liver was homogenized in 350 µl buffer (10 mM HEPES-KOH, pH 7.6, 1.5 mM MgCl$_2$, 10 mM KCl, 5 mM EDTA, 5 mM EGTA, and 250 mM sucrose) supplemented with a protease inhibitor cocktail consisting of 0.1 mM leupeptin, 5 mM dithiothreitol, 1 mM PMSF, 0.5 mM Pefabloc, 5 µg/ml pepstatin A, 25 µg/ml N-acetyl-leu-leu-norleucinal, and 10 µg/ml aprotinin. The homogenates were then passed through a 22-gauge needle 10–15 times and subjected to centrifugation at 1000 X g for 5 min at 4°C. The 1000 X g pellet was resuspended in 500 µl of buffer (20 mM HEPES-KOH, pH 7.6, 2.5% (v/v) glycerol, 0.42 M NaCl, 1.5 mM MgCl$_2$, 1 mM EDTA, 1 mM EGTA) supplemented with the protease inhibitor cocktail, rotated for 30 min at 4°C, and centrifuged at 100,000 X g for 30 min at 4°C. The supernatant from this spin was precipitated with 1.5 ml cold acetone at −20°C for at least 30 min; the precipitated material was collected by centrifugation, resuspended in SDS-lysis buffer (10 mM Tris-HCl, pH 6.8, 1% (w/v) SDS, 100 mM NaCl, 1 mM EDTA, and 1 mM EGTA), and designated the nuclear extract fraction. The post-nuclear supernatant from the original spin was used to prepare the membrane fraction by centrifugation at 100,000 X g for 30 min at 4°C. Each membrane fraction was resuspended in 100 µl SDS-lysis buffer.

Protein concentration of nuclear extract and membrane fractions were measured using the BCA Kit (ThermoFisher Scientific). Prior to SDS-PAGE, aliquots of the nuclear extract fractions were mixed with 5X SDS-PAGE loading buffer to achieve a final concentration of 1X. Aliquots of the membrane

fractions were mixed with an equal volume of buffer containing 62.5 mM Tris-HCl, pH 6.8, 15% (w/v) SDS, 8 M urea, 10% (v/v) glycerol, and 100 mM DTT, after which 5X SDS loading buffer was added to a final concentration of 1X. Nuclear extract fractions were boiled for 5 min, and membrane fractions were incubated for 20 min at 37°C prior to SDS-PAGE. After SDS-PAGE, proteins were transferred to Hybond C-Extra nitrocellulose filters (GE Healthcare, Piscataway, NJ). The filters were incubated with the antibodies described below and in the figure legends. Bound antibodies were visualized with peroxidase-conjugated, affinity-purified donkey anti-mouse or anti-rabbit IgG (Jackson ImmunoResearch Laboratories, Inc, West Grove, PA) using the SuperSignal CL-HRP substrate system (ThermoFisher Scientific) according to the manufacturer's instructions. Gels were calibrated with prestained molecular mass markers (Bio-Rad, Hercules, CA). Filters were exposed to film at room temperature. Antibodies used for immunoblotting to detect mouse SREBP-1 (rabbit polyclonal IgG-20B12), SREBP-2 (rabbit monoclonal IgG-22D5), Insig-1 (rabbit polyclonal anti-Insig-1 antiserum), HMGCR (rabbit polyclonal IgG-839c and mouse monoclonal IgG-A9), UBXD8 (rabbit polyclonal IgG-819), and Scap (IgG-R139) were previously described (*Jo et al., 2011*; *Engelking et al., 2005*; *McFarlane et al., 2014*). Rabbit polyclonal IgG-205 was raised against amino acids 2–21 of mouse UBIAD1. Rabbit polyclonal IgG-492 was raised against a C-terminal sequence (CKVIPEKSHQE) of hamster Insig-2. Rabbit polyclonal anti-calnexin IgG was purchased from Novus Biologicals (Littleton, CO). Rabbit polyclonal anti-LSD1 IgG was obtained from Cell Signaling (Beverly, MA). Rabbit polyclonal anti-GM130 was obtained from Abcam (Cambridge, MA). All antibodies were used at a final concentration of 1–5 µg/ml; the anti-Insig-1 antiserum was used at a dilution of 1:1000.

To obtain hepatic Golgi- and ER-enriched membrane fractions, approximately 150 mg of frozen livers were homogenized in buffer containing 50 mM Tris-HCl (pH 7.5), 150 mM NaCl, the protease inhibitor cocktail, and 15% (w/v) sucrose. Following homogenization using a ball-bearing homogenizer with a 10 µm clearance, the samples were centrifuged at 3000 X g for 10 min. The resulting post nuclear supernatants (PNS) were then applied to a discontinuos sucrose gradient that was generated and centrifuged at 110,000 X g as described previously (*Radhakrishnan et al., 2008*). Golgi- and ER-enriched fractions were collected and subjected to immunoblot analysis as described in figure legends.

Corneas from *Ubiad1*[WT/WT] and *Ubiad1*[Ki/Ki] mice were collected and fixed in 10% neutral-buffered formalin, followed by paraffin embedding and sectioning. The sections were analyzed by immunohistochemistry with polyclonal IgG-839c (against HMGCR) as described previously (*Evers et al., 2010*).

## Lipid Analysis

Cholesterol and sterol biosynthetic intermediates were measured using LC-MS/MS according to the method of McDonald et al. (*McDonald et al., 2012*; *Mitsche et al., 2015*). Briefly, sterols were isolated on an LC gradient (Shimadzu LC-20) and detected using the MRM pair on a triple quadrapole MS (ABSciex 4000 q-TRAP) and quantified against authentic sterol standards (Avanti Polar Lipids, Alabaster, AL).

Nonsterol isoprenoids were measured as follows. Approximately 100 mg of tissue was homogenized using a BeadRuptor (Omni International, Kennesaw, GA). Samples were placed in a mixture of 1 mL 1:1 MeOH:IPA containing 2 ng each of d5-GGOH and d8-MK-4 using 2 mL beadruptor tubes. The slurry was transferred to glass 16 × 100 mm glass tubes with PTFE-lined screw-caps. The resulting tissue slurry was sonicated for 5 min. A 5 mL aliquot of acetone was added, and samples were vortexed for three minutes followed by incubation at room temperature for 5 min. The vortex and incubation steps were repeated twice for a total of three cycles. Samples were then centrifuged at 3500 rpm for 5 min at 4°C. Supernatant was removed to a new 16 × 100 mm tube. The pellet was washed with 1 mL of acetone, centrifuged at 3500 rpm for 5 min at 4°C, and the resultant supernatant was pooled with the first extract. The sample was dried under nitrogen at a temperature of approximately 35°C. The dried extract was dissolved in 5 mL hexane, after which 2 mL of water was added and the sample was placed in a freezer at −20°C until the lower layer was frozen. The hexane layer was removed from the frozen aqueous phase using a Pasteur pipette. When the aqueous phase thawed, 2 mL hexane was added, and the process was repeated with the two hexane extracts being pooled. The sample was dried under nitrogen at ~35°C. The dried extract was dissolved in 400 µL hexane, transferred to a 2 mL autosampler vial with low-volume insert and dried again. The final sample was dissolved in 400 µL 90% methanol that had been warmed to ~37°C.

Analytes were measured using Shimadzu LC-20XR high-performance liquid-chromatography (HPLC) coupled to a SCIX API 5000 triple quadrupole mass spectrometer (Shimadzu, Columbia

Maryland; SCIEX, Framingham, MA). A 10 µL sample was injected onto an Agilent Poroshel EC-C$_{18}$ (Agilent, Santa Clara, CA; 150 × 2.1 mm, 2.7 µm) and resolved with a ternary gradient using A 90% methanol, B methanol, and C dichloromethane (DCM). A and B also contained 0.1% acetic acid. The gradient began at 100% A, increased to 100% B over 2 min, held at 100% B for 0.25 min, then increased to 3:7 (B:C) over 6.75 min. The column was re-equilibrated first with 100% B for 2 min, then 100% A for 1 min. Flow rate was 0.5 mL/min and the column was maintained at 35°C. Analytes were ionized using atmospheric pressure chemical ionization (APCI) at 400°C and detected using multiple reaction monitoring (MRM).

Masses and other MS parameters are given below :

|  | Q1 | Q3 | DP | CE | Source |
|---|---|---|---|---|---|
| GGOH | 273 | 71 | 77 | 48 | Sigma |
| MK-4 | 445.5 | 187 | 59 | 40 | Cerilliant |
| PK | 451.3 | 187 | 59 | 40 | Cerilliant |
| d8-MK4 | 452.4 | 94 | 59 | 50 | Sigma (catalog #737836) |
| MK7 | 649.5 | 187 | 59 | 40 | Cerilliant |
| CoQ-10 | 863.7 | 197 | 59 | 44 | Cerilliant |
| d5-GGOH | 278.3 | 81 | 77 | 48 | Avanti Polar Lipids (custom-synthesized) |

Reproducibility of data – All results were confirmed in at least two independent experiments conducted on different days using different animals and batches of cells.

## Acknowledgements

We thank Lisa Beatty and Ijeoma Onweneme for help with tissue culture and Guosheng Liang for helpful advice. We also thank Dr. Jerry Niederkorn for help in the analysis of *Ubiad1*$^{Ki/Ki}$ for corneal opacification. This work was supported by National Institutes of Health grants HL-20948 and GM-112409 to RD-B.

## Additional information

### Funding

| Funder | Grant reference number | Author |
|---|---|---|
| National Institutes of Health | HL-20948 | Russell A DeBose-Boyd |
| National Institutes of Health | GM-112409 | Russell A DeBose-Boyd |

The funders had no role in study design, data collection and interpretation, or the decision to submit the work for publication.

### Author contributions

Youngah Jo, Conceptualization, Data curation, Formal analysis, Validation, Investigation, Visualization, Methodology, Writing—original draft, Writing—review and editing; Jason S Hamilton, Data curation, Validation, Investigation, Visualization, Methodology; Seonghwan Hwang, Resources, Validation, Investigation, Methodology; Kristina Garland, Gennipher A Smith, Shan Su, Iris Fuentes, Sudha Neelam, Bonne M Thompson, Jeffrey G McDonald, Investigation, Methodology; Russell A DeBose-Boyd, Conceptualization, Supervision, Funding acquisition, Visualization, Writing—original draft, Project administration, Writing—review and editing

### Author ORCIDs

Jason S Hamilton (iD) http://orcid.org/0000-0002-3291-3125
Shan Su (iD) http://orcid.org/0000-0002-9369-1642
Russell A DeBose-Boyd (iD) http://orcid.org/0000-0002-7452-5227

## Ethics

Animal experimentation: This study was performed in strict accordance with the recommendations in the Guide for Care and Use of Laboratory Animals of the National Institutes of Health. All of the animal work described in this manuscript has been approved and conducted under the oversight of the UT Southwestern Institutional Animal Care and Use Committee (Protocol # - 2016-101636).

## Decision letter and Author response

Decision letter https://doi.org/10.7554/eLife.44396.019
Author response https://doi.org/10.7554/eLife.44396.020

# Additional files

## Supplementary files

• Transparent reporting form
DOI: https://doi.org/10.7554/eLife.44396.016

## Data availability

All data generated in this study are included in the manuscript.

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
