## [Decision Letter]

Thank you for submitting your article "Schnyder corneal dystrophy-associated UBIAD1 Inhibits ER-associated degradation of HMG CoA reductase in mice" for consideration by *eLife*. Your article has been reviewed by three peer reviewers, one of whom is a member of our Board of Reviewing Editors, and the evaluation has been overseen by Randy Schekman as the Senior Editor. The following individual involved in review of your submission has agreed to reveal his identity: Randy Y Hampton (Reviewer #3).

The reviewers have discussed the reviews with one another and the Reviewing Editor has drafted this decision to help you prepare a revised submission.

The referees all agreed that the quality of the data is high and that the generation and initial characterization of a mouse model for SCD warrants publication as a Research Advance linked to your prior paper. They further judged that the study provides important in vivo support for prior mechanistic models on the link between UBIAD1, HMG CoA reductase, and cholesterol homeostasis established from cell culture. One referee raised a number of potential additional mechanistic links that might be important to explore. During our discussion, we decided that these extensions of the project are beyond the scope of the current paper. Although we have appended the full referee comments for your interest, we expect you to address only the minor suggestions for improvement provided below. anticipate that addressing these comments will probably only need clarifications to the text, perhaps some re-organisation of figure panels, and possibly answering queries on unclear points.

*Reviewer #1:*

This Research Advance contribution from the Debose-Boyd lab follows up an earlier study from this group analysing the function of UBIAD1 as it relates to HMGCR regulation. The earlier study reported that UBIAD1 stimulates HMGCR degradation in a ppGG-dependent manner, and that dominant human SCD disease mutants in UBIAD1 interfere with full HMGCR degradation under cholesterol-replete conditions. The hypothesis from the earlier work was that the disease mutants may work by causing elevated HMGCR, resulting in elevated cholesterol which eventually accumulates in the eye to cause corneal opacities. The current study examines this idea in mice by generating a knock-in of one such SCD disease mutant (N100S). Analysis of the mouse reveals clear evidence of dysregulated HMGCR including its elevated steady state levels, various compensatory sequelae, and elevated levels of sterol-related molecules. Importantly, the mouse appears to also show age-dependent opacities in the eye. Analysis of MEFs from the mutant mice verified altered cholesterol-mediated HMGCR degradation. Considered together with their earlier work, the study makes a strong case that the cholesterol-containing deposits observed in SCD result from altered HMGCR regulation caused by its excessively strong interaction with mutant UBIAD1. It is important because it provides a nice mouse model for SCD, and more speculatively, provides additional in vivo insights into the intricacies of cholesterol homeostasis. The paper is generally well organized and written.

1) As I am sure the authors are aware, another group recently published a preliminary analysis of a N100S UBIAD1 knock-in mouse (PMID 29977031). This earlier study lacks the molecular analysis of HMGCR, age-dependent phenotypes, and mechanistic interpretations of the current work. Nevertheless, it is important that the authors provide a careful discussion of these earlier findings and how they relate to the current study.

2) In Figure 1B, it would be useful to show some indicator of equal loading (perhaps the stained blot showing total protein or a blot for an unrelated protein). In the graph, it would be useful to have error bars on the protein levels if available. I realise the next figure provides a more thorough analysis with suitable controls, but a reader is immediately faced with a rather sparse and somewhat incomplete observation right at the outset. Perhaps one option is to simply omit Figure 1 and start with Figure 2 since this is better characterized anyway.

3) Figure 1B would be easier to follow in the text and legend if the three sub-panels were labelled as panels B, C, and D. This would make it easier to call out the individual experiments. The same applies to other figure panels where more than one type of experiment is shown together (e.g., the top and bottom parts of Figure 2A. In general, it is easier for the reader if each panel displays a single experiment.

4) The increase in UBIAD1 in Ki livers (and other tissues) is based on increased levels in the membrane fraction. Is it possible that the total levels are the same, and the difference is due to efficiency of recovery of certain subcellular membranes? Figure 6 shows that mutant UBIAD1 is preferentially in the ER whereas wild type UBIAD1 is preferentially in the Golgi. So if the ER is more effectively recovered than Golgi in the membrane fraction, then one might get the impression that total UBIAD1 is increased. Can a blot also be done on total lysate to clarify this issue? Total lysate is shown in Figure 6C, but it is unclear if the two blots are directly comparable to infer anything about total UBIAD1 levels.

5) I think it is more useful for a reader to directly state the absolute level of increase in HMGCR rather than the value after normalization to mRNA levels. The absolute level of protein is what is important for its enzymatic activity, which is the key parameter for the disease (i.e., the amount of cholesterol that accumulates) that forms the focus of this paper. The normalized values inform more about the relative rates of synthesis/degradation, which is also interesting but less central to the phenotype. Thus, one could simply report total values in the figures and Results section, then in the discussion state how the levels are increased despite much lower mRNA levels, suggesting that either synthesis rate is higher and/or degradation rate is lower.

6) Results section – It states that the Lovastatin effect on HMGCR is blunted in Ki mice liver relative to wild type, but the result in Figure 6A shows that accumulation of HMGCR is enhanced in Ki mice liver. Perhaps the sentence is worded incorrectly?

7) Results section – It says Lovastatin led to a slight elevation of HMGCR in Ki mice; However, it was not easy for me to judge from the blot if the increase was any more or less than in wild type mice. Perhaps it is useful to quantify the result and be more precise in the wording with something like "led to XX-fold elevation above the already high levels of HMGCR in Ki mice…" to emphasize that the levels at baseline are already very high.

8) It is worth stating how frequently (if at all) corneal inclusions are observed in wild type and heterozygous mice of similar age. This is important to make sure that the inclusions are not simply a consequence of old age.

*Reviewer #2:*

This manuscript follows up on earlier work from the deBose-Boyd laboratory on the regulation of HMG-CoA-reductase (HMGCR) by UBIAD1, a protein mutated in Schnyder Corneal Dystrophy (SCD) patients. Here a SCD-associated UBIAD1 mutation (N100S) knock-in mouse (Ubiad1ki/ki) is described. It is convincingly shown that Ubiad1ki/ki mouse recapitulates several aspects of SCD pathology, including corneal opacification in aged mice. Analysis of tissues and mouse embryonic fibroblast (MEF) derived from the knock-in mouse largely confirm UBIAD1 effects on HMGCR levels described earlier in cell lines and further characterize its consequences on SREBP transcription program. The Ubiad1ki/ki mouse model appears a useful tool and may prime further research into SCD disease mechanism and treatment. However, I am not convinced that at this stage this work provides a significant advance on the HMGCR regulation by UBIAD1 or on SCD pathophysiology. This study touches on several questions but leaves them mostly unanswered. For example:

1) In Ubiad1ki/ki mice, both HMGCR and UBIAD1 protein levels are increased, however no mechanistic insight is provided to explain these observations, particularly the increase in UBIAD1 protein levels by the N100S mutation.

2) Previous work indicate that UBIAD1 inhibits HMGCR degradation by acting in a post ubiquitination state. This study does not clarify the mechanism by which UBIAD1 prevents HMGCR degradation and how this is impacted by the N100S mutation. Does this mutant bind tighter to HMGCR or the effects are simply a result of its higher protein levels? It is assumed that its lower capacity to bind to GGPP is responsible for tighter binding to HMGCR but this is not tested.

3) In Ubiad1ki/ki mice, HMGCR levels are disturbed in all tissues analyzed. However, it is unclear why the disease manifest specifically in the eye. Is HMGCR more susceptible to UBIAD1 mediated control in the eye while other mechanisms dominate in the remaining organs? Or does the overall impact of UBIAD1 in cholesterol homeostasis insufficient to lead to other complications such as atherosclerosis?

4) Why have the authors chosen to backcross their C57BL/6 x 129 Ubiad1ki/ki mice to C57BL/6 pure genetic backgrounds? This may be not be trivial for non-experts.

5) The quality of the SREBP-2 blots are poor throughout the manuscript.

Figure 2A: Poor quality SREBP-2 blots.

Figure 5A; Lane 5 and 6 seem to be reversed regarding the SREBP-2 protein levels both in membrane fraction and nuclear extract. While the authors provided some explanation for this in the main text, it would be more convincing if the authors could comment on this more clearly or repeat the western blot.

Figure 5C; Poor quality of SREBP-2 blot and the SREBP-2 blotting for membrane fraction is missing.

6) Overexposed HMGCR blots

Figure 4C; While it is expected that HMGCR protein level are lower in WT and specifically increased in Ubiad1ki/ki mouse tissues (and MEF), due to overexpression, it is difficult to assess if this is the case in the HMGCR blot in Figure 4C.

Figure 6A; The authors claim that lovastatin caused HMGCR accumulation in Ubiad1ki/ki mouse liver. However, this is difficult to assess when the HMGCR blot is overexposed and no quantification of protein levels is provided.

7) In Figure 7D, the author showed there is no significant increase of total cholesterol in the range of 300-400ng sterol/ug DNA and similar total cholesterol measurement in Figure 7E (for measurement of intermediates of cholesterol synthesis) presented in the range of 100-200ng sterol/ug DNA.

8) Textual changes

Last paragraph, last line should refer to Figure 7E, not 7D.

The legend of Figure 4C and 4D is confusing. Figure references do not seem to be correct. Also, the use of MG132 is mentioned, but nowhere in the figures it is annotated. Please amend.

*Reviewer #3:*

The manuscript "Schnyder corneal dystrophy-associated UBIAD1 Inhibits ER-associated degradation of HMG CoA reductase in mice" is a beautiful addition to the regulated ERAD and sterol physiology literature, with the thoroughness and quality that I have come to expect from Dr. Debose-Boyd and his intrepid group. It is very pleasing to see such a thoughtful and well-rendered test of a disease model; this work most certainly should be published in *eLife*.

The only thing I am not totally clear on is the experiment in Figure 5C. In that experiment, the highly stabilized K→R form of HMGCR (shoot, my whole career I have called it HMGR…) is used as an excellent pre-stabilized control. But if UBIAD only works by allowing enhanced degradation of HMGCR, why are the effects of UBIAD mutant on wild-type HMGCR, which presumably still undergo some degradation, more pronounced than the degradation behavior of the cis highly stabilized mutant?

If this were one of those more evil journals that care not about a PIs resources, or the time of their hard working underlings, I would suggest an experiment with the double mutant, to see if there are regulatory effects that transcend control of degradation. Fortunately, the *eLife* ethos (which I enthusiastically endorse after spending 50,000 dollars for an extraneous, confirmatory biochemical test of an air-tight genetic study (true story)) encourages reviewers to find merit in the extant work and resist requesting more experiments. So, maybe an explanation or even a mention of this feature of the Figure 5C results would be useful.

Another question that might be interesting to explore in this brief study is the connection (or lack thereto) between this opaquitopathy (I just made that up; not bad!)-meaning opacity-causing disease- and the recent work on cataracts that appear to be relieved by lanosterol, since HMGCR would be expected to promote synthesis of more of this alleviating agent in the disease modeled herein. Different mechanism? Different pathology? No relation? Just an interesting thing to reference in the discussion maybe. Or maybe not.

Otherwise, full speed ahead. Fantastic work!

---

## [Author Response]

Reviewer #1:

[…] The paper is generally well organized and written.1) As I am sure the authors are aware, another group recently published a preliminary analysis of a N100S UBIAD1 knock-in mouse (PMID 29977031). This earlier study lacks the molecular analysis of HMGCR, age-dependent phenotypes, and mechanistic interpretations of the current work. Nevertheless, it is important that the authors provide a careful discussion of these earlier findings and how they relate to the current study.

We are aware that another group published a preliminary analysis of UBIAD1 (N100S) knockin mice, which lacks molecular characterization of HMGCR and documentation of age-dependent opacification of knockin corneas. The authors report evidence of mitochondrial damage and accumulation of glycerophosphoglycerol in corneas of the knockin mice. Based on this finding, the authors propose that mitochondrial dysfunction is linked to the abnormal cholesterol deposition that occurs in SCD patients. We have now included a discussion of these results in the revised manuscript.

2) In Figure 1B, it would be useful to show some indicator of equal loading (perhaps the stained blot showing total protein or a blot for an unrelated protein). In the graph, it would be useful to have error bars on the protein levels if available. I realise the next figure provides a more thorough analysis with suitable controls, but a reader is immediately faced with a rather sparse and somewhat incomplete observation right at the outset. Perhaps one option is to simply omit Figure 1 and start with Figure 2 since this is better characterized anyway.3) Figure 1B would be easier to follow in the text and legend if the three sub-panels were labelled as panels B, C, and D. This would make it easier to call out the individual experiments. The same applies to other figure panels where more than one type of experiment is shown together (e.g., the top and bottom parts of Figure 2A. In general, it is easier for the reader if each panel displays a single experiment.

It is important to note that although separated into different panels, membrane and nuclear extract fractions immunoblotted in Figure 1B were obtained from the same group of livers. We noted in the figure legend that “although shown in separate panels, calnexin and LSD-1 serves as loading controls for the HMGCR and nuclear SREBP immunoblots.” The remainder of experiments (e.g., Figure 2A) were conducted similarly and the fact that loading controls are shown in separate panels was indicated in the figure legend. To improve clarity of Figure 1B, we have removed the HMGCR mRNA data (which appears in Figure 1—figure supplement 1A) and estimation of the relative amount of HMGCR protein. We now report the absolute level of HMGCR protein under the immunoblot as suggested by this reviewer (point 5). We have also indicated in the figure legend that “although shown in a separate panel, LSD-1 serves as a loading control for the nuclear SREBP immunoblots.”

4) The increase in UBIAD1 in Ki livers (and other tissues) is based on increased levels in the membrane fraction. Is it possible that the total levels are the same, and the difference is due to efficiency of recovery of certain subcellular membranes? Figure 6 shows that mutant UBIAD1 is preferentially in the ER whereas wild type UBIAD1 is preferentially in the Golgi. So if the ER is more effectively recovered than Golgi in the membrane fraction, then one might get the impression that total UBIAD1 is increased. Can a blot also be done on total lysate to clarify this issue? Total lysate is shown in Figure 6C, but it is unclear if the two blots are directly comparable to infer anything about total UBIAD1 levels.

Throughout the studies presented in this manuscript, we have immunoblotted membrane fractions for ERlocalized calnexin or UBXD8 (Figure 3) to control for loading. Even though levels of UBIAD1 and HMGCR accumulate in livers and other tissues of knockin mice, the levels of calnexin and UBXD8 remain constant. If the ER was more efficiently recovered than Golgi in the membrane fractions of the knockin mice, it would be anticipated that levels of both calnexin and UBXD8 would also be increased.

5) I think it is more useful for a reader to directly state the absolute level of increase in HMGCR rather than the value after normalization to mRNA levels. The absolute level of protein is what is important for its enzymatic activity, which is the key parameter for the disease (i.e., the amount of cholesterol that accumulates) that forms the focus of this paper. The normalized values inform more about the relative rates of synthesis/degradation, which is also interesting but less central to the phenotype. Thus, one could simply report total values in the figures and Results section, then in the discussion state how the levels are increased despite much lower mRNA levels, suggesting that either synthesis rate is higher and/or degradation rate is lower.

We now report the absolute levels of HMGCR rather than values after normalization. These values are included under the HMGCR immunoblots in Figure 1B and Figure 2B.

6) Results section – It states that the Lovastatin effect on HMGCR is blunted in Ki mice liver relative to wild type, but the result in Figure 6A shows that accumulation of HMGCR is enhanced in Ki mice liver. Perhaps the sentence is worded incorrectly?

The sentence to which the reviewer refers is confusing and has been removed.

7) Results section – It says Lovastatin led to a slight elevation of HMGCR in Ki mice; However, it was not easy for me to judge from the blot if the increase was any more or less than in wild type mice. Perhaps it is useful to quantify the result and be more precise in the wording with something like "led to XX-fold elevation above the already high levels of HMGCR in Ki mice…" to emphasize that the levels at baseline are already very high.

We have removed the sentence that states lovastatin caused a slight increase in the amount of HMGCR protein in eyes of the knockin mice.

8) It is worth stating how frequently (if at all) corneal inclusions are observed in wild type and heterozygous mice of similar age. This is important to make sure that the inclusions are not simply a consequence of old age.

We have now indicated in the Results section that none of the wild type UBIAD1 (N100S) knockin mice developed corneal opacification and the heterozygous animals were not examined.

Reviewer #2:

[…] This study touches on several questions but leaves them mostly unanswered. For example:1) In Ubiad1ki/ki mice, both HMGCR and UBIAD1 protein levels are increased, however no mechanistic insight is provided to explain these observations, particularly the increase in UBIAD1 protein levels by the N100S mutation.2) Previous work indicates that UBIAD1 inhibits HMGCR degradation by acting in a post ubiquitination state. This study does not clarify the mechanism by which UBIAD1 prevents HMGCR degradation and how this is impacted by the N100S mutation. Does this mutant bind tighter to HMGCR or the effects are simply a result of its higher protein levels? It is assumed that its lower capacity to bind to GGPP is responsible for tighter binding to HMGCR but this is not tested.3) In Ubiad1ki/ki mice, HMGCR levels are disturbed in all tissues analyzed. However, it is unclear why the disease manifest specifically in the eye. Is HMGCR more susceptible to UBIAD1 mediated control in the eye while other mechanisms dominate in the remaining organs? Or does the overall impact of UBIAD1 in cholesterol homeostasis insufficient to lead to other complications such as atherosclerosis?4) Why have the authors chosen to backcross their C57BL/6 x 129 Ubiad1ki/ki mice to C57BL/6 pure genetic backgrounds? This may be not be trivial for non-experts.

We backcrossed the knockin mice to pure C57BL/6 genetic background to ensure observed phenotypes are not influenced by the mixed genetic background. This rationale is now included in the revised manuscript.

5) The quality of the SREBP-2 blots are poor throughout the manuscript.Figure 2A: Poor quality SREBP-2 blots.Figure 5A; Lane 5 and 6 seem to be reversed regarding the SREBP-2 protein levels both in membrane fraction and nuclear extract. While the authors provided some explanation for this in the main text, it would be more convincing if the authors could comment on this more clearly or repeat the western blot.Figure 5C; Poor quality of SREBP-2 blot and the SREBP-2 blotting for membrane fraction is missing.

In the revised manuscript, we have provided darker exposures of the nuclear SREBP-2 blots for Figure 2A and Figure 5C. We have observed some variation in the quality of SREBP-2 nuclear immunoblots over the years and do not understand the basis for this. It should be noted; however, that the immunoblots show a decrease in nuclear SREBP-2 in UBIAD1 (N100S) knockin mice fed chow or cholesterol-containing diets, which correlates with reduction in mRNAs encoding its target genes. In Figure 5C, we only immunoblotted for nuclear SREBP-2 as a control for cholesterol feeding. Figure 5 now includes mRNA data showing that cholesterol feeding downregulates mRNAs encoding SREBP target genes in livers of both wild type and knockin mice.

Lanes 5 and 6 of Figure 5A were not reversed. The result shows that cholesterol feeding causes an increase in the precursor of SREBP-2, which likely results from inhibition of its cleavage to the mature nuclear form.

6) Overexposed HMGCR blotsFigure 4C; While it is expected that HMGCR protein level are lower in WT and specifically increased in Ubiad1ki/ki mouse tissues (and MEF), due to overexpression, it is difficult to assess if this is the case in the HMGCR blot in Figure 4C.Figure 6A; The authors claim that lovastatin caused HMGCR accumulation in Ubiad1ki/ki mouse liver. However, this is difficult to assess when the HMGCR blot is overexposed and no quantification of protein levels is provided.

Figure 4C shows an experiment in which MEFs were first depleted of isoprenoids through incubation in medium containing lipoprotein-deficient serum and the HMGCR inhibitor compactin. As a result of this depletion, HMGCR transcription, translation, and stability are increased. These increases result in the accumulation of HMGCR protein to levels that approach those observed in the knockin MEFs. In the isoprenoid-replete conditions shown in Figure 4A, levels of HMGCR protein are low because of sufficient levels of sterol and nonsterol isoprenoids; HMGCR protein accumulates in Ubiad1^Ki^ cells because of its resistance to degradation.

7) In Figure 7D, the author showed there is no significant increase of total cholesterol in the range of 300-400ng sterol/ug DNA and similar total cholesterol measurement in Figure 7E (for measurement of intermediates of cholesterol synthesis) presented in the range of 100-200ng sterol/ug DNA.

Corneas from different sets of mice were analyzed in Figure 7E and 7D, which may explain differences in total cholesterol levels.

8) Textual changesLast paragraph, last line should refer to Figure 7E, not 7D.The legend of Figure 4C and 4D is confusing. Figure references do not seem to be correct. Also, the use of MG132 is mentioned, but nowhere in the figures it is annotated. Please amend.

We now changed Figure 7D to Figure 7E in the last paragraph as indicated by this reviewer.

The reviewer is correct, the legend to Figure 4 is wrong. We should have indicated that all of the cells in D received MG-132. This has now been corrected in the revised manuscript.

Reviewer #3:

[…] Another question that might be interesting to explore in this brief study is the connection (or lack thereto) between this opaquitopathy (I just made that up; not bad!)-meaning opacity-causing disease- and the recent work on cataracts that appear to be relieved by lanosterol, since HMGCR would be expected to promote synthesis of more of this alleviating agent in the disease modeled herein. Different mechanism? Different pathology? No relation? Just an interesting thing to reference in the discussion maybe. Or maybe not.Otherwise, full speed ahead. Fantastic work!

We thank the reviewer for the supportive comments!